# Distinct regulatory ribosomal ubiquitylation events are reversible and hierarchically organized

Danielle M Garshott, Elayanambi Sundaramoorthy, Marilyn Leonard, Eric J Bennett*

Section of Cell and Developmental Biology, Division of Biological Sciences, University of California, San Diego, La Jolla, United States

**Abstract** Activation of the integrated stress response (ISR) or the ribosome-associated quality control (RQC) pathway stimulates regulatory ribosomal ubiquitylation (RRub) on distinct 40S ribosomal proteins, yet the cellular role and fate of ubiquitylated proteins remain unclear. We demonstrate that uS10 and uS5 ubiquitylation are dependent upon eS10 or uS3 ubiquitylation, respectively, suggesting that a hierarchical relationship exists among RRub events establishing a ubiquitin code on ribosomes. We show that stress dependent RRub events diminish after initial stimuli and that demodification by deubiquitylating enzymes contributes to reduced RRub levels during stress recovery. Utilizing an optical RQC reporter we identify OTUD3 and USP21 as deubiquitylating enzymes that antagonize ZNF598-mediated 40S ubiquitylation and can limit RQC activation. Critically, cells lacking USP21 or OTUD3 have altered RQC activity and delayed eS10 deubiquitylation indicating a functional role for deubiquitylating enzymes within the RQC pathway.

*For correspondence:
e1bennett@ucsd.edu

**Competing interests:** The authors declare that no competing interests exist.

## Introduction

The proteome must continuously adapt to changing environmental conditions and exposure to extrinsic proteotoxic stressors that challenge cellular, tissue, and organismal health. A prominent source of proteotoxic stress arises during translation where transcriptional or mRNA processing errors can result in the translation of defective or truncated proteins and lead to the accumulation of toxic nascent protein products (*Brandman and Hegde, 2016*; *Schuller and Green, 2018*). Failure to remove these deleterious proteins can lead to aggregation and contribute to human pathologies including a wide range of neurodegenerative disorders (*Gestwicki and Garza, 2012*). A variety of cellular quality control and stress response pathways have evolved to guard against the accumulation of these aberrant nascent polypeptides and maintain cellular homeostasis (*Dubnikov et al., 2017*; *Lykke-Andersen and Bennett, 2014*; *Sontag et al., 2017*). One prominent example is the integrated stress response (ISR) which is activated by a variety of protein homeostasis stressors. ISR activation results in rapid global protein synthesis attenuation while also stimulating the translation of critical stress response factors, including protein chaperones and ubiquitin ligases, that assist in rebalancing homeostasis (*Guan et al., 2017*; *Pakos-Zebrucka et al., 2016*). Quality control pathways safeguard against the accumulation of potentially toxic misfolded or otherwise aberrant proteins. The ribosome-associated quality control (RQC) pathway is one such quality control system that identifies elongating ribosomal complexes whose progression is halted due to a defect in the translating mRNA or emerging nascent chain (*Brandman and Hegde, 2016*). After the initial recognition event, RQC pathway components catalyze the degradation of both the mRNA and nascent polypeptide, followed by ribosome subunit recycling (*Ikeuchi et al., 2018*). Defects within the RQC pathway result in the production of aberrant protein products and an eventual accumulation of protein aggregates (*Choe et al., 2016*; *Defenouillère et al., 2016*; *Yonashiro et al., 2016*).

**eLife digest** Ribosomes are cellular machines that build proteins by latching on and then reading template molecules known as mRNAs. Several ribosomes may be moving along the same piece of mRNA at the same time, each making their own copy of the same protein. Damage to an mRNA or other problems may cause a ribosome to stall, leading to subsequent collisions.

A quality control pathway exists to identify stalled ribosomes and fix the 'traffic jams'. It relies on enzymes that tag halted ribosomes with molecules known as ubiquitin. The cell then removes these ribosomes from the mRNA and destroys the proteins they were making. Afterwards, additional enzymes take off the ubiquitin tags so the cell can recycle the ribosomes. These enzymes are key to signaling the end of the quality control event, yet their identity was still unclear.

Garshott et al. used genetic approaches to study traffic jams of ribosomes in mammalian cells. The experiments showed that cells added sets of ubiquitin tags to stalled ribosomes in a specific order. Two enzymes, known as USP21 and OTUD3, could stop this process; this allowed ribosomes to carry on reading mRNA. Further work revealed that the ribosomes in cells that produce higher levels of USP21 and OTUD3 were less likely to stall on mRNA. On the other hand, ribosomes in cells lacking USP1 and OTUD3 retained their ubiquitin tags for longer and were more likely to stall.

The findings of Garshott et al. reveal that USP21 and OTUD3 are involved in the quality control pathway which fixes ribosome traffic jams. In mice, problems in this pathway have been linked with neurons dying or being damaged because toxic protein products start to accumulate in cells; this is similar to what happens in human conditions such as Alzheimer's and Parkinson's diseases. Using ubiquitin to target and potentially fix the pathway could therefore open the door to new therapies.

Protein ubiquitylation plays a key role during these stress response and quality control pathways to facilitate the degradation of misfolded or damaged proteins (*Bengtson and Joazeiro, 2010*; *Brandman et al., 2012*; *Joazeiro, 2017*; *Joazeiro, 2019*; *Pilla et al., 2017*). Monoubiquitylation, which typically does not target proteins for degradation, of distinct ribosomal proteins is also stimulated in response to ISR activation and conditions that stimulate RQC suggesting that ubiquitylation regulates these pathways beyond protein degradation (*Garzia et al., 2017*; *Higgins et al., 2015*; *Ikeuchi et al., 2019*; *Juszkiewicz et al., 2018*; *Matsuo et al., 2017*; *Simms et al., 2017*; *Sugiyama et al., 2019*; *Sundaramoorthy et al., 2017*). Studies in both *S. cerevisiae* and mammalian systems have identified a list of RQC factors and have delineated a series of events that occur when ribosome progression is slowed enough to initiate a QC response (*Joazeiro, 2019*). Regulatory ribosomal ubiquitylation (RRub) has emerged as a conserved critical initiating signal during RQC events (*Ikeuchi et al., 2019*; *Juszkiewicz and Hegde, 2017*; *Matsuo et al., 2017*; *Simms et al., 2017*; *Sundaramoorthy et al., 2017*). In mammals, the ubiquitin ligase ZNF598 catalyzes site-specific ubiquitylation of eS10 (RPS10) and uS10 (RPS20) to resolve ribosomes that have stalled during decoding of polyA sequences (*Juszkiewicz and Hegde, 2017*; *Sundaramoorthy et al., 2017*). Ablation of ZNF598 or the ribosomal protein RACK1, as well as conserved ubiquitylated target lysines in uS10 or eS10 results in RQC failure and subsequent readthrough of stall inducing sequences (*Juszkiewicz and Hegde, 2017*; *Sundaramoorthy et al., 2017*). Similar, yet distinct ubiquitylation events regulate RQC in yeast (*Matsuo et al., 2017*). Current models suggest that ribosome collisions are the key initiation signal which recruits critical ubiquitin ligases to facilitate RRub allowing for subsequent nascent chain ubiquitylation, mRNA degradation, and ribosome recycling (*Ikeuchi et al., 2019*; *Juszkiewicz et al., 2018*; *Simms et al., 2017*). The observation that both uS10 and eS10 ubiquitylation are required for mammalian RQC suggest a potential structured order of ubiquitylation events may be needed to specifically mark collided ribosomes. While it is clear that RRub is required for downstream RQC events, the precise mechanistic role the 40S ubiquitylation plays during RQC and the consequence of ubiquitylation on target ribosomal proteins remain open questions.

Activation of the integrated stress response (ISR) in mammalian cells triggers an additional set of RRub events on uS3 (RPS3) and uS5 (RPS2) that do not require ZNF598 and do not function within the RQC pathway and whose function remains uncharacterized (*Higgins et al., 2015*). The presence of two separate ubiquitylation events on neighboring ribosomal proteins again suggests a possible

hierarchical relationship among distinct RRub events that likely impart separate functions. Studies in mammalian cells have demonstrated that the extent of ISR-stimulated uS3 and uS5 monoubiquitylation diminished upon removal of ISR agonists (*Higgins et al., 2015*). This observation suggests that either RRub events are reversed by the action of deubiquitylating enzymes (Dubs) or that ubiquitin-modified ribosomal proteins are degraded after RQC events.

Here, we establish that regulatory ribosomal ubiquitylation events are reversible and mediated by deubiquitylating enzymes following activation of the ISR or RQC pathways. We utilized an overexpression screen to identify two Dubs, USP21 and OTUD3, whose expression stimulates readthrough of poly(A)-mediated ribosome stalls. We demonstrate that USP21 and OTUD3 can directly antagonize ZNF598-mediated eS10 and uS10 ubiquitylation events. Further, we show that USP21 and OTUD3 expression results in augmented removal of ubiquitin from eS10 and uS10 following UV-induced RQC. USP21 expression also represses ISR-stimulated uS3 and uS5 ubiquitylation. Importantly, cells lacking USP21 or OTUD3 display reduced levels of poly(A)-mediated stall readthrough and a delay in eS10 demodification following UV-induced RQC activation. Expression of OTUD3 results in enhanced stall readthrough compared to knock-in cell lines engineered to lack either eS10 or uS10 RRub sites indicating that combinatorial ribosomal ubiquitylation is required for optimal RQC function. Interestingly, we demonstrate that uS10 ubiquitylation is dependent upon eS10 ubiquitylation and that uS5 ubiquitylation requires uS3 ubiquitylation further suggesting a hierarchical relationship upon RRub events. Taken together, our results establish that RRub events are reversible by deubiquitylating enzymes and that RRub represents a combinational post-translational code that imparts distinct functional outcomes on ribosomes.

## Results

### Regulatory ribosomal ubiquitylation is reversible

Previous studies demonstrated that the integrated stress response (ISR)-stimulated regulatory ubiquitylation of uS5 (RPS2) and uS3 (RPS3) is diminished upon cessation of the ISR (*Higgins et al., 2015*). The reduced levels of ubiquitylated uS5 or uS3 after ISR stimulation could be the result of demodification by a deubiquitylating enzyme (Dub) or turnover of ribosomal proteins. We also reasoned that the ZNF598-catalyzed ubiquitylation of uS10 (RPS20) and eS10 (RPS10), a critical initiating signal within the ribosome associated quality control (RQC) pathway, may also be reversible. To examine the reversibility of RRub events, HCT116 cells were treated with the translation elongation inhibitor anisomycin (ANS) to induce both ribosome stalling, as well as inhibit global protein synthesis (*Juszkiewicz et al., 2018*). This allowed us to simultaneously observe the timing of RRub demodification relative to total protein turnover in the absence of new protein synthesis. Consistent with previous studies, ANS induced ubiquitylation of both eS10 and uS10 (*Figure 1A*; *Juszkiewicz et al., 2018*; *Matsuo et al., 2017*). While eS10 ubiquitylation diminished and uS10 ubiquitylation persisted over time, there was no discernable reduction in the relative amount of the unmodified form of each protein (*Figure 1A,B*). Further no accumulation of unmodified or ubiquitylated eS10 or uS10 was observed with combined treatment of proteasome inhibitors and ANS over 12 hr indicating that ribosomal proteins are either not rapidly degraded when ribosome stalling is stimulated, or degradation cannot be detected within the limit of immunoblotting approaches used in these experiments. To address whether deubiquitylation is observed with uS5 and uS3 RRub events, we transiently exposed HCT116 cells with the ISR agonist dithiotheritol (DTT) alone or in combination with the protein synthesis inhibitor cycloheximide (CHX) followed by DTT washout with and without cycloheximide. DTT stimulated uS5 and uS3 ubiquitylation which subsequently diminished to pre-treatment levels over time (*Figure 1C*). Further, the relative amount of the unmodified protein remained stable despite global protein synthesis attenuation (*Figure 1C,D*). The varying kinetics of demodification observed with CHX treatment as compared to DTT alone correlates with heightened ISR activation and prolonged phosphorylation of eIF2α which results in sustained uS3 and uS5 ubiquitylation. Together these results suggest that the loss in the amount of ubiquitylated ribosomal proteins likely results from demodification by deubiquitylating enzymes.

To further examine the reversibility and timing of distinct RRub events, we utilized UV-induced ISR activation which stimulates uS5, uS3, eS10 and uS10 regulatory ubiquitylation (*Elia et al., 2015*; *Higgins et al., 2015*). We exposed 293T cells to UV and allowed cells to recover for increasing

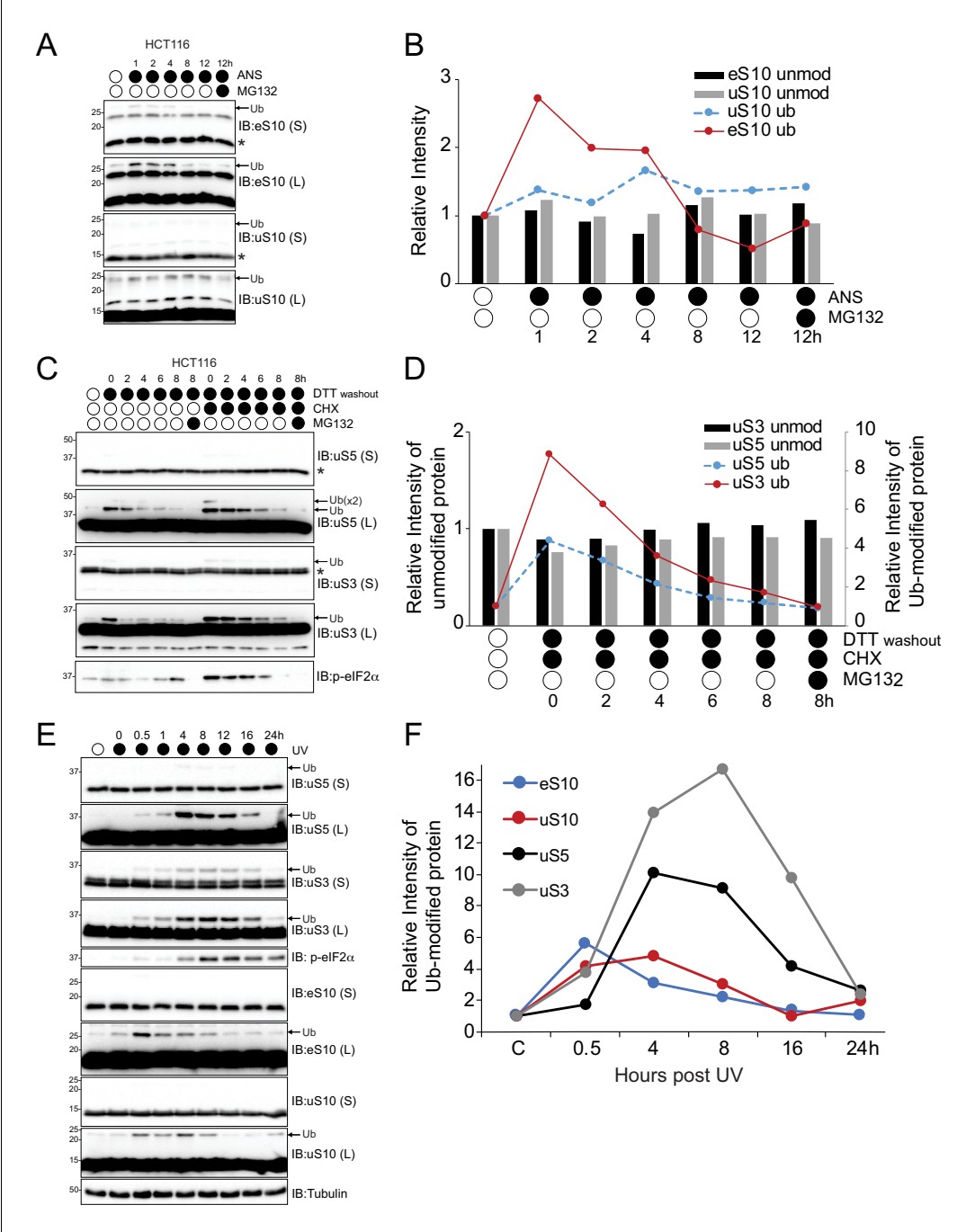

**Figure 1.** Stress-induced RRub events are reversible. (**A**) HCT116 cells were treated with anisomycin (ANS, 5 μg/ml) alone or with MG132 (10 μM) for the indicated times. Whole-cell extracts were analyzed by SDS-PAGE and immunoblotted with the indicated antibodies. The ubiquitin-modified and unmodified ribosomal protein is indicated by the arrow and asterisk, respectively. S and L denote short and long exposures (n = 1). (**B**) The amount of ubiquitylated eS10 (red line) and uS10 (blue dashed line) and unmodified eS10 (black column) and uS10 (gray column) after the indicated treatments compared to untreated cells quantified from panel A. (**C**) HCT116 cells were treated with DTT (5 mM) alone or with cycloheximide (CHX, 100 μg/ml) for 4 hr followed by DTT washout in media with or without CHX alone or with MG132 (10 μM) for the indicated times. Whole-cell extracts were analyzed by SDS-PAGE and immunoblotted with the indicated antibodies. The ubiquitin-modified and unmodified ribosomal protein is indicated by the arrow and asterisk, respectively. S and L denote short and long exposures (n = 1). (**D**) The amount of ubiquitylated uS3 (red line) and uS5 (blue dashed line) and unmodified uS3 (black column) and uS5 (gray column) after the indicated treatments compared to untreated cells quantified from panel C. (**E**) 293 T cells were exposed to UV and allowed to recover for the indicated times. Whole-cell extracts were analyzed by SDS-PAGE and immunoblotted with the indicated antibodies. The ubiquitin-modified ribosomal protein is indicated by the arrow. S and L denote short and long exposures (n = 1). (**F**) The

*Figure 1 continued on next page*

Figure 1 continued

amount of ubiquitylated eS10 (blue line), uS10 (red line), uS5 (black line) and uS3 (gray line) after UV exposure compared to untreated cells quantified from panel E.

periods of time. All observed RRub events diminished over time further indicating the RRub events are reversible (*Figure 1E,F*). Interestingly, eS10 and uS10 ubiquitylation preceded uS5 and uS3 ubiquitylation after UV exposure (*Figure 1E,F*). Consistent with previous studies demonstrating that ISR-stimulated uS5 and uS3 RRub events require eIF2α phosphorylation, uS5 and uS3 ubiquitylation occurred coincidently with eIF2α phosphorylation (*Higgins et al., 2015*). This timing offset between RRub events and the demonstration that uS10 and eS10 ubiquitylation is catalyzed by ZNF598 whereas uS5 and uS3 ubiquitylation does not require ZNF598 suggests that uS5 and uS3 RRub events are functionally distinct from the more immediate eS10 and uS10 ubiquitylation that likely occur as a direct result of UV-induced ribosomal stalls.

## Distinct sets of RRub events are hierarchically organized

To investigate the importance of individual RRub events, we generated point mutant knock-in HCT116 cell lines in which the endogenous eS10, uS10, uS3 or uS5 loci were modified by CRISPR/Cas9 approaches to replace previously identified ubiquitylated lysine residues with arginine. We first examined if mutating RRub lysine residues resulted in altered protein stability. We observed no appreciable change in total protein abundance in the eS10 K138R/K139R knock-in (eS10-KI) and uS10 K4R/K8R knock-in (uS10-KI) cells following ANS treatment (*Figure 2A,B* and *Figure 2—figure supplement 1A,B*). Similarly, the inability to ubiquitylate uS5 K54R/K58R (uS5-KI) and uS3 K214R (uS3-KI) did not change the steady-state abundance or the turnover of uS5 or uS3 (*Figure 2C,D* and *Figure 2—figure supplement 1C,D*). Interestingly in the course of this experiment and validation of these cell lines we noticed a hierarchal relationship among the ubiquitylation events (*Figure 2—figure supplement 1E,F*). As expected, eS10-KI cell lines completely lack ANS-induced eS10 ubiquitylation. However, loss of eS10 ubiquitylation substantially reduces uS10 ubiquitylation (*Figure 2A* and *Figure 2—figure supplement 1F*) as compared to control cell lines. The inability to ubiquitylate eS10 had a negligible impact on the levels of UV-induced uS5 and uS3 ubiquitylation (*Figure 2—figure supplement 1F*). In contrast, uS10-KI cells maintained their ability to ubiquitylate eS10, uS5, and uS3 despite the expected loss of uS10 ubiquitylation (*Figure 2B* and *Figure 2—figure supplement 1E*). These results indicate that eS10 ubiquitylation may be required for optimal uS10 ubiquitylation upon induction of RQC events. Similar to the hierarchy of eS10 and uS10, the lack of DTT-induced uS3 ubiquitylation in the uS3-KI cells results in complete ablation of uS5 modification while loss of uS5 ubiquitylation did not effect uS3 DTT-stimulated RRub (*Figure 2C,D*). Combined, these results suggest that hierarchical relationships exist within distinct classes of RRub events and imply a specific order of ubiquitylation events.

## Identification of deubiquitylating enzymes that antagonize eS10 and uS10 regulatory ubiquitylation

Our results implicate the direct involvement of deubiquitylating enzymes in regulating RRub and RQC function. To identify and characterize deubiquitylating enzymes (Dubs) that operate within the RQC pathway, we utilized a previously established dual-fluorescence RQC reporter assay in which a stall-inducing poly(A) sequence placed between GFP and cherry fluorescent protein (ChFP) coding sequences results in the repression of downstream ChFP fluorescence as compared to GFP, indicative of an RQC event initiating upon translation of the poly(A) sequence. (*Juszkiewicz and Hegde, 2017*; *Sundaramoorthy et al., 2017*). Previous studies demonstrated that loss of ZNF598 function and the resulting decrease in eS10 and uS10 ubiquitylation results in readthrough of poly(A) sequences and a subsequent increase in the ChFP:GFP ratio of the stall reporter (*Juszkiewicz and Hegde, 2017*; *Sundaramoorthy et al., 2017*). Overexpression of a deubiquitylating enzyme that mediates deubiquitylation of eS10 and uS10 RRub events would phenocopy ZNF598 loss-of-function and enhance the amount of poly(A) readthrough. Based on this rationale, a panel of 60 human Dub expression plasmids were individually co-transfected with the poly(A) stall reporter plasmid into 293T cells and the corresponding ChFP:GFP ratio was measured by flow cytometry. Immunoblotting

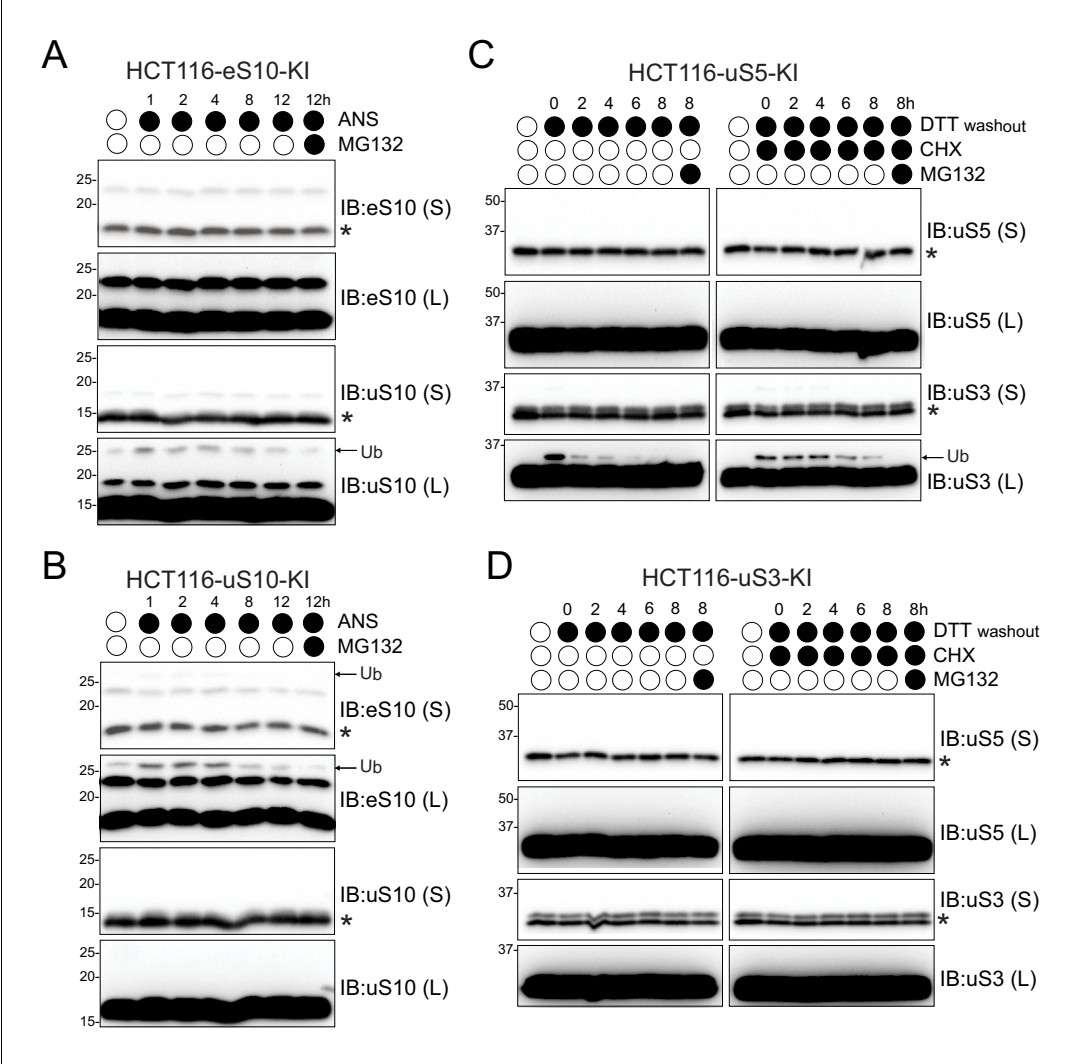

**Figure 2.** Distinct sets of RRub events are hierarchically organized. (A,B) HCT116 mutant RRub knock-in (KI) cell lines eS10-KI (K138RK139R) or uS10-KI (K4RK8R) (A,B) were treated with ANS (5 µg/ml) and MG132 (10 µM) continuously for the indicated times. Whole-cell extracts were analyzed by SDS-PAGE and immunoblotted with the indicated antibodies. (C,D) HCT116 mutant uS5-KI (K54RK58R) or uS3-KI (K214R) were either untreated or pretreated with DTT (5 mM) alone or with CHX (100 µg/ml) for 4 hr followed by DTT washout in media with or without CHX (100 µg/ml) and MG132 (10 µM). Cells were collected at the indicated times post DTT washout. Whole-cell extracts were analyzed by SDS-PAGE and immunoblotted with the indicated antibodies. The ubiquitin-modified and unmodified ribosomal protein is indicated by the arrow and asterisk, respectively. S and L denote short and long exposures (n = 1).

The online version of this article includes the following figure supplement(s) for figure 2:

**Figure supplement 1.** Quantification of site-specific RRub demodification upon exposure to stress.

confirmed the expression of 58 Dubs albeit to varying expression levels (*Figure 3—figure supplement 1*). A Z-score analysis of the ChFP:GFP ratio for the stall reporter identified six candidate Dubs whose expression resulted in the largest enhancement of poly(A) readthrough above the population mean (*Figure 3A*). We validated that expression of the six candidate Dubs resulted in a reproducible enhancement of poly(A)-stall readthrough and a subsequent increase in ChFP:GFP values using the stall-reporter assay (*Figure 3B*). To directly validate that the resulting increase in the ChFP:GFP ratio was specific to the poly(A) reporter, each candidate Dub was expressed with a control plasmid lacking the internal poly(A) sequence (*Figure 3B*). OTUB2, OTUD3, USP10 and UCHL1 expression did not alter the ChFP:GFP ratio of the control reporter while OTUD1 and USP21 only modestly elevated the ChFP:GFP ratio indicating that the identified Dubs specifically alter the ability of ribosomes to progress through a poly(A)-induced ribosomal stall (*Figure 3B*).

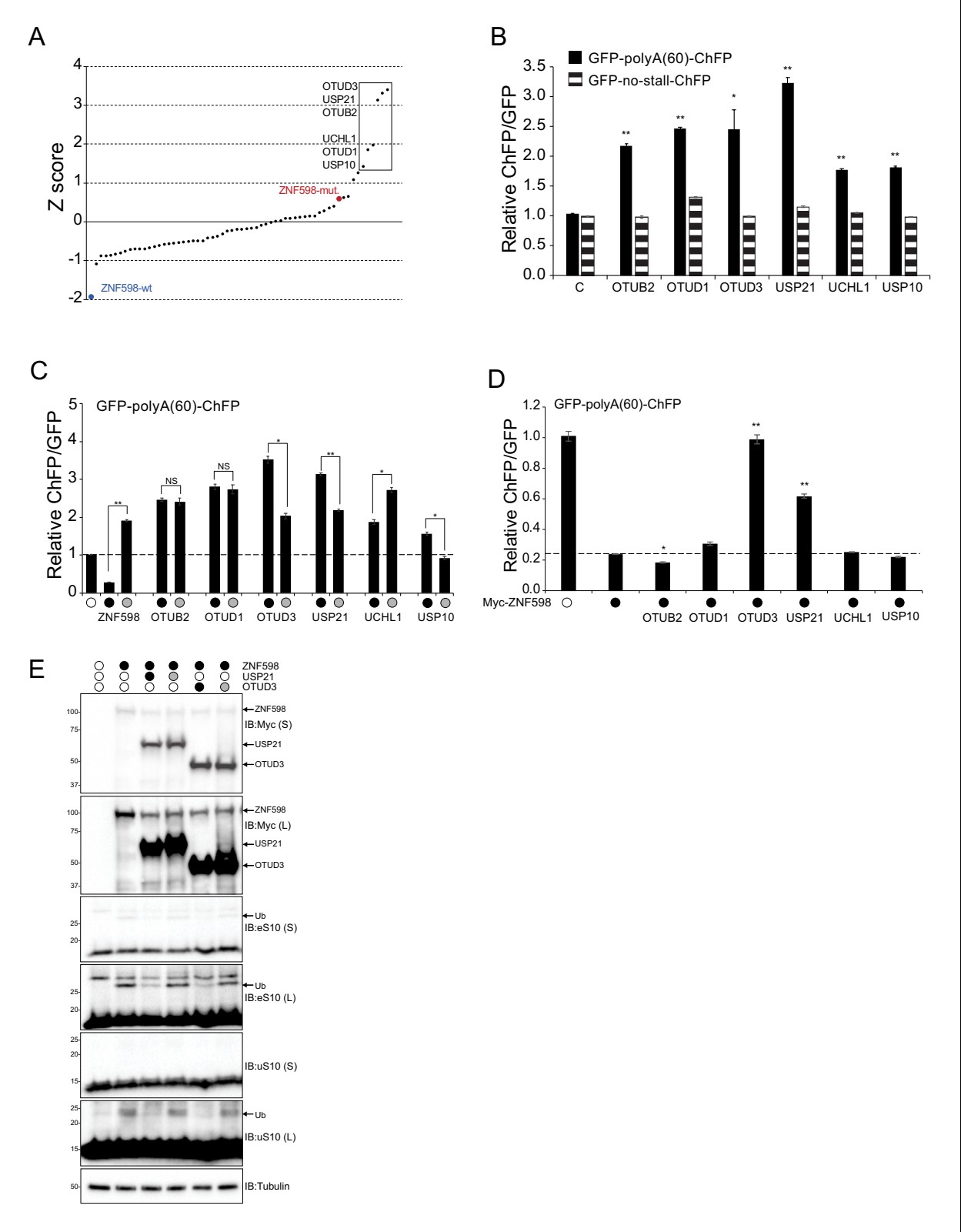

**Figure 3.** Identification of deubiquitylating enzymes that allow for readthrough of poly(A)-mediated ribosome stalls. (A) 60 human deubiquitylating enzyme (Dub) expression plasmids were individually co-transfected with the poly(A)-stall reporter plasmid and the resulting ChFP and GFP fluorescence intensities were measured by flow cytometry. The Z-score value for each Dub is depicted (n = 1). The six Dubs with the highest Z-score are boxed to indicate candidate Dubs that induce increased readthrough of poly(A)-stall reporter. Expression of wild type (blue dot) and catalytically inactive (red

*Figure 3 continued on next page*

*Figure 3 continued*

dot) ZNF598 are shown as controls. (**B**) The ChFP:GFP ratio from cells transfected with the poly(A)-stall reporter (solid bars) or a reporter containing no stall sequence (striped bars), along with expression plasmids for the indicated Dubs relative to a control plasmid. Error bars denote SEM for triplicate transfections. \*\*$p<0.0001$, \*$p<0.01$ using Student's t-test comparing Dub expression to control transfections. (**C**) The ChFP:GFP ratio from cells transfected with expression plasmids for either wild type (black circle) or catalytically inactive (gray circle) Dubs and the poly(A)-stall reporter relative to a control plasmid. Control transfections with the poly(A) reporter plasmid alone are indicated by the open circle. Error bars denote SEM for triplicate transfections. \*\*$p<0.0001$, \*$p<0.001$ using Student's t-test comparing the wild type to the catalytically inactive mutant for each Dub or ZNF598. NS = not significant. (**D**) The ChFP:GFP ratio from cells transfected with the poly(A)-stall reporter plasmid and ZNF598 alone or in combination with the indicated wild type Dubs (black circle) relative to a control plasmid. Control transfections with the poly(A) reporter plasmid alone are indicated by the open circle. Error bars denote SEM for triplicate transfections. \*\*$p<0.0001$, \*$p<0.01$, using Student's t-test comparing wild type ZNF598 alone to samples with ZNF598 and the indicated Dub. (**E**) Whole-cell extracts from ZNF598- knockout (KO) cells transfected with expression plasmids for wild type (black circles) ZNF598, USP21, or OTUD3 and their respective inactive mutants (gray circles) were analyzed by SDS-PAGE and immunoblotted using the indicated antibodies. The ubiquitin-modified ribosomal protein is indicated by the arrow. S and L denote short and long exposures (n = 1). The online version of this article includes the following figure supplement(s) for figure 3:

**Figure supplement 1.** Validation of human Dub expression plasmids.
**Figure supplement 2.** OTUD3 and USP21 enhance poly(A)-stall readthrough in a non-synergistic and ZNF598 dependent manner.

## Overexpression of candidate RQC-Dubs results in poly(A) stall-sequence readthrough in an activity-dependent manner

To examine whether the overexpression-induced increase in poly(A)-mediated stall readthrough was dependent on the catalytic activity of each of the identified Dubs, catalytically inactive versions of each Dub were generated by site-directed mutagenesis to convert the critical catalytic cysteine residue to serine. Each Dub and the respective catalytically inactive mutant (CS) were co-expressed with the poly(A)-stall reporter (*Figure 3C*). Expression of OTUB2 and OTUD1 inactive mutants resulted in an equivalent degree of poly(A)-stall readthrough as compared to the respective wild type enzymes (*Figure 3C*). Additionally, expression of inactive UCHL1 resulted in enhanced readthrough of the poly(A)-sequence compared to wild type UCHL1. This result suggests that the observed increase in ChFP fluorescence does not require the catalytic activity of OTUB2 or OTUD1. In contrast, expression of the inactive mutants for the deubiquitylating enzymes USP21, OTUD3, and USP10 resulted in a substantial reduction of the ChFP:GFP ratios compared to wild type versions. Expression of inactive OTUD3 or USP21 resulted in elevated stall readthrough compared to control transfections indicating a possible alternative role for OTUD3 and USP21 within the RQC pathway that is ZNF598 independent. Together, these results indicate that USP21, OTUD3, and USP10 expression results in elevated poly(A)-mediated stall readthrough in an activity-dependent manner.

## USP21 and OTUD3 antagonize ZNF598-mediated RRub events

Having demonstrated that exogenous expression of USP21, OTUD3, and USP10 enhanced poly(A) stall-induced readthrough, we wanted to examine the ability of the Dubs to directly antagonize ZNF598-mediated translational stalling of the poly(A) reporter. As expected, exogenous expression of wild type ZNF598 resulted in decreased ChFP:GFP ratios as compared to control transfections (*Figure 3D*). Next, the poly(A) reporter and candidate Dubs were expressed along with exogenous ZNF598. Both USP21 and OTUD3, when co-expressed with ZNF598, resulted in a greater than 2.5 fold increase in the ChFP:GFP poly(A) reporter ratio relative to what was observed when ZNF598 was expressed in isolation (*Figure 3D*). Antagonism with ZNF598 was not observed for USP10 which suggests its role within the RQC operates independently of ZNF598. Combined expression of OTUD3 and USP21 did not further enhance poly(A)-mediated stall readthrough events or result in enhanced antagonism of ZNF598 (*Figure 3—figure supplement 2A*). These results are consistent with the hypothesis that USP21 and OTUD3 directly antagonize ZNF598-mediated RRub events. Immunoblot analysis revealed that cells solely overexpressing ZNF598 displayed a 5-fold increase in the abundance of ubiquitylated eS10 compared to untransfected cells (*Figure 3E*). Exogenous expression of USP21 substantially reduced the ZNF598-stimulated eS10 and uS10 ubiquitylation in an activity-dependent manner (*Figure 3E*). The same result was observed upon expression of OTUD3 and ZNF598 (*Figure 3E*). These results demonstrate the ability of USP21 and OTUD3 to remove ubiquitin from eS10 and uS10 following ZNF598-mediated RRub events. Because USP21 and

OTUD3 were the only Dubs to show activity dependent antagonism of ZNF598 in our stall read-through assay, these Dubs were selected for subsequent analyses.

To further examine the antagonism between ZNF598 and OTUD3 and USP21, parental HCT116 cells and ZNF598 knockout (ZNF598-KO) cells were transfected with either USP21 or OTUD3 expressing plasmids and the poly(A) stall-inducing reporter. We reasoned that expression of these Dubs in the absence of ZNF598 would not impact the amount of poly(A)-stall readthrough beyond that observed with loss of ZNF598 expression. As expected, expression of either Dub along with the poly(A)-reporter in parental HCT116 cells markedly increased the ChFP:GFP ratio of the stall reporter (*Figure 4A,B*). ZNF598-KO cells displayed the expected elevated ChFP:GFP ratio of the stall reporter which was modestly enhanced upon further expression of either Dub (*Figure 4A,B*). This modest enhancement was also observed upon expression of either the wild type or inactive ver-sions of other Dubs, OTUB2, OTUD1, and UCHL1 in the ZNF598-KO cell line suggesting that the increased readthrough is possibly non-specific and does not require Dub activity (*Figure 3—figure supplement 2B*). However, the observation that USP21 or OTUD3 expression modestly augments readthrough of poly(A) sequences in cells lacking ZNF598 suggests that USP21 and OTUD3 may function within the RQC in a ZNF598 independent manner while also directly antagonizing ZNF598 ribosomal ubiquitylation.

## USP21 and OTUD3 deubiquitylate ZNF598 substrates eS10 and uS10

To investigate the role of individual RRub events during RQC, we utilized the uS10-KI and eS10-KI HCT116 cell lines to examine if the enhanced readthrough of poly(A) stall-inducing sequences observed upon USP21 or OTUD3 overexpression required uS10 or eS10 ubiquitylation. Consistent with our previous results, eS10-KI and uS10-KI cell lines allowed for enhanced readthrough of poly(A)-mediated stall events using our stall reporter FACS assay whereas uS3-KI and uS5-KI cell lines did not appreciably alter reporter levels compared to parental cells (*Figure 2—figure supplement 1G*; *Sundaramoorthy et al., 2017*). We expressed OTUD3 or USP21 in uS10 or eS10 knock-in cell lines along with the poly(A) stall reporter. USP21 or OTUD3 overexpression in either eS10-KI or uS10-KI cells resulted in a further enhancement of the ChFP:GFP ratio above the respective transfec-tion controls (*Figure 4C,D*). This enhancement was largely activity-dependent as expression of inac-tive OTUD3 reduced the extent of readthrough compared to wild type in both eS10 and uS10 knock-in cell lines (*Figure 4D*). Expression of inactive USP21 resulted in reduced readthrough com-pared to wild type in uS10-KI cells but not eS10-KI cells (*Figure 4C*). These results indicate that OTUD3 and USP21 can demodify both uS10 and eS10, consistent with our immunoblotting data (*Figure 3E*). Further, these results indicate that the combined loss of uS10 and eS10 RRub events results in a stronger RQC defect than loss of either uS10 or eS10 ubiquitylation events alone.

To validate the poly(A)-reporter results, we immunoblotted cell lysates in which we expressed either wild type or inactive USP21 or OTUD3 in parental or eS10-KI or uS10-KI cell lines to visualize eS10 and uS10 ubiquitylation. As expected, ZNF598 expression stimulated eS10 and uS10 ubiquity-lation in parental HCT116 cells (*Figure 4E*). ZNF598 expression in uS10-KI cells failed to induce uS10 ubiquitylation without impacting the ability of ZNF598 to ubiquitylate eS10. Conversely, ZNF598 expression failed to ubiquitylate eS10 in eS10-KI cells, and uS10 ubiquitylation was substantially reduced compared to ZNF598 expression in parental cells. This is consistent with a model in which eS10 ubiquitylation is needed prior to uS10 ubiquitylation. While expression of wild type USP21 or OTUD3 reduced the abundance of both monoubiquitylated eS10 and uS10 in parental HCT116 cells, expression of the inactive variants restored ubiquitylation to steady-state levels (*Figure 4E*). Expres-sion of either Dub in the eS10-KI cell line could further demodify the small amount of uS10 ubiquity-lation observed upon ZNF598 expression (*Figure 4E*). Similarly, USP21 and OTUD3 antagonized the ZNF598-dependent eS10 ubiquitylation in uS10-KI cells in an activity dependent manner. Taken together, these results indicate that USP21 or OTUD3 can deubiquitylate both eS10 and uS10, resulting in enhanced readthrough of poly(A)-mediated stall events. Further, these results demon-strate that the combined ubiquitylation of eS10 and uS10 is required for optimal resolution of RQC events.

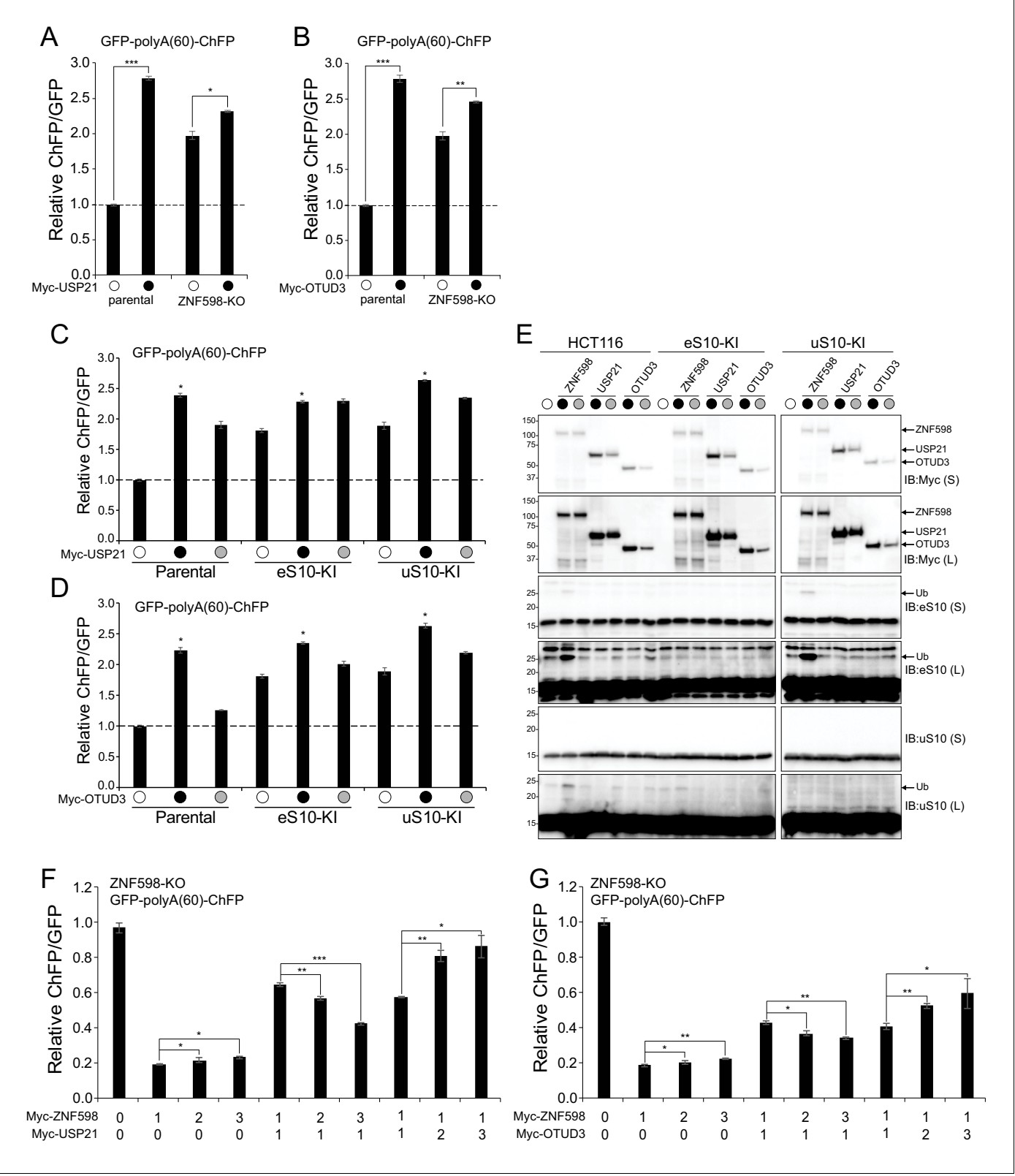

**Figure 4.** USP21 and OTUD3 antagonize ZNF598-mediated RRub events. (**A,B**) Parental HCT116 cells and ZNF598 knock-out (KO) cells were transfected with USP21 (**A**) or OTUD3 (**B**) expression plasmids and the poly(A)-stall reporter (black circles) or the poly(A)-stall reporter alone (open circles). Fluorescence intensities were measured by flow cytometry and the relative ChFP:GFP ratio is depicted. Error bars denote SEM for triplicate transfections. ***p<0.0001, **p<0.001, *p<0.05, using Student's t-test comparing Dubs to control transfection. (**C,D**) The ChFP:GFP ratio from parental

*Figure 4 continued on next page*

*Figure 4 continued*

HCT116 cells or point mutant knock-in (KI) eS10 or uS10 cell lines transfected with the poly(A)-stall reporter alone (open circles) or with expression plasmids for wild type (black circles) or inactive mutant (gray circles) USP21 (C) or OTUD3 (D) relative to control transfections in the indicated cell lines. Error bars denote SEM for triplicate transfections. *p<0.0001 using Student's t-test comparing wild type Dub transfections to control transfection in the indicated cell lines. (E) Whole-cell extracts from cells transfected as indicated in panels C and D were analyzed by SDS-PAGE and immunoblotted for the indicated antibodies. Black and gray circles denote expression of wild type or inactive versions, respectively. The ubiquitin-modified ribosomal protein is indicated by the arrow. S and L denote short and long exposures (n = 1). (F,G) The ChFP:GFP ratio from HCT116 ZNF598 knockout (KO) cells transfected with increasing amounts of plasmid DNA for either wild type ZNF598 and USP21 (F) or OTUD3 (G) and the poly(A)-stall reporter. Numbers indicate the ratio of transfected DNA for each plasmid. Error bars represent SEM of triplicate replicates. ***p<0.0001, **p<0.001, *p<0.05 using Student's t-test comparing the different ZNF598 to Dub DNA ratios as indicated.

## The abundance of ZNF598 in relation to USP21 or OTUD3 governs RQC events

Examination of quantitative proteomic datasets from human cell lines revealed that ZNF598 protein levels are 19-fold in excess of OTUD3 while USP21 levels were undetectable indicating that ZNF598 protein levels are in vast excess of either RRub Dub at steady state (*Itzhak et al., 2016*). Given the relative excess of ZNF598 compared to its antagonizing Dubs, we set out to examine how varying the levels of the Dubs relative to ZNF598 would impact RQC events. We transfected increasing amounts of a ZNF598 expressing plasmid in ZNF598-KO cell lines and examined poly(A)-mediated stall readthrough events using the stall reporter assay. Expression of the poly(A)-reporter with increasing concentrations of exogenous ZNF598 in isolation did not result in a dose-dependent decrease in the ChFP:GFP ratio suggesting that ZNF598 expression at the lowest levels tested were sufficient to fully restore RQC function and that elevated ZNF598 levels do not further enhance ribosome stall resolution (*Figure 4F,G* and *Figure 3—figure supplement 2C,D*). We then varied the relative ZNF598 expression levels compared to either USP21 or OTUD3 and examined the impact on the ChFP:GFP ratio of the stall reporter. When equal amounts of USP21 or OTUD3 and ZNF598 expression plasmids were transfected, a substantial increase in the ChFP:GFP ratio was observed as compared to ZNF598 expressed alone which further verified the direct antagonism observed previously (*Figure 4F,G* and *Figure 3—figure supplement 2C,D*). The reporter ChFP:GFP ratio declined as ZNF598 plasmid transfections were doubled and tripled with respect to USP21 or OTUD3 plasmid DNA (*Figure 4F,G* and *Figure 3—figure supplement 2C,D*). Conversely, doubling and tripling the expression of USP21 while holding ZNF598 expression levels constant revealed additional readthrough of the poly(A) stall-inducing sequence with increasing ChFP:GFP ratios, suggesting further antagonism of the ligase. This result suggests that maintaining ZNF598 expression levels high relative to USP21 and OTUD3 is a feature that may be required to enable rapid 40S ribosomal ubiquitylation upon RQC-triggering events that are not immediately removed by antagonistic Dubs. These results also indicate that OTUD3 or USP21 overexpression (40-fold and 100-fold above endogenous, respectively) is required to compete with endogenous ZNF598 activity.

## OTUD3 and USP21 deubiquitylate 40S ribosomal proteins following RRub induction

To examine the ability of USP21 and OTUD3 to catalyze deubiquitylation of RRub events, we generated doxycycline (Dox)-inducible 293 cell lines that conditionally express either the wild type or inactive version of each Dub. To observe the reversal of RRub events, we induced ribosome stalling and subsequent RRub using UV exposure. To test the impact of Dub expression, cells were either treated with or without Dox for 16 hr prior to UV exposure. Cells were then UV irradiated and allowed to recover for increasing periods of time. Based on our previously established reversibility of RRub events, we suspected that overexpression of wild type USP21 or OTUD3 would induce a more rapid removal of eS10 and uS10 ubiquitylation during recovery from UV-induced stress. Control cells without induction of Dub overexpression displayed induced eS10 and uS10 ubiquitylation immediately following UV treatment, followed by rapid demodification 4 hr after UV exposure (*Figure 5A–D* and *Figure 5—figure supplement 1*). Interestingly, in cells overexpressing exogenous wild type USP21, the amount of detectable eS10 ubiquitylation rapidly declined 1 hr post UV exposure, while uS10 ubiquitylation was completely ablated (*Figure 5A* and *Figure 5—figure supplement 1*). These observations suggest that USP21 can demodify eS10 and uS10 following UV-induced stress. To

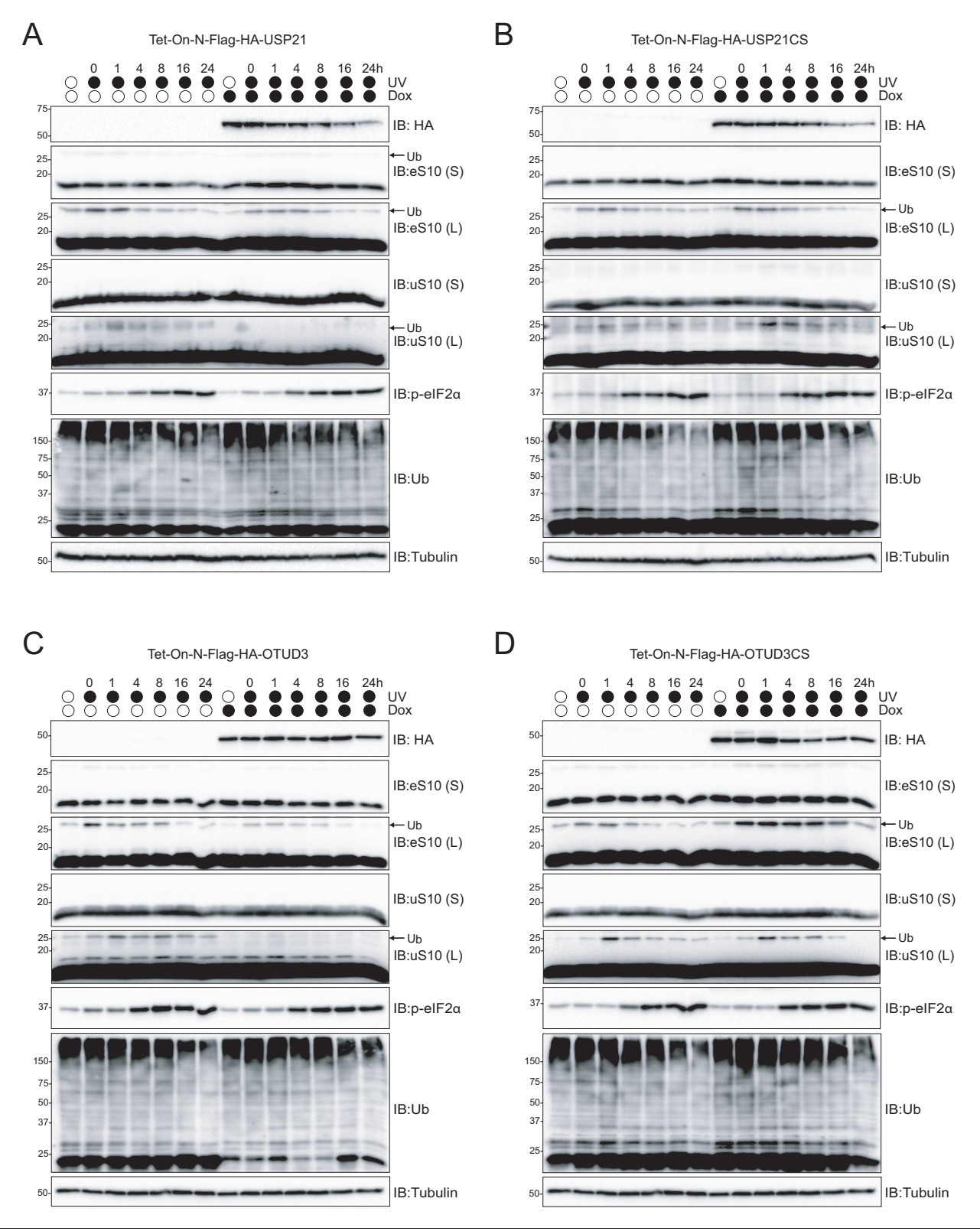

**Figure 5.** USP21 and OTUD3 expression accelerates RRub demodification following UV exposure. (A–D) Cells with stable inducible expression of wild type USP21 (A), OTUD3 (C) or inactive versions (B, D, respectively) were induced or uninduced with doxycycline (Dox, 2 μg/ml) for 16 hr followed by UV exposure. Whole cell extracts from cells collected at the indicated times after UV exposure were analyzed by SDS-PAGE and immunoblotted using the indicated antibodies. The ubiquitin-modified ribosomal protein is indicated by the arrow. S and L denote short and long exposures (n = 1).

*Figure 5 continued on next page*

*Figure 5 continued*

The online version of this article includes the following figure supplement(s) for figure 5:

**Figure supplement 1.** Quantification of site-specific RRub demodification upon Dub overexpression.

substantiate these observations, we induced the expression of inactive USP21 and determined the dynamicity of UV-induced RRub events. Immunoblots confirmed that the dynamics of eS10 and uS10 ubiquitylation following UV treatment in cells expressing inactive USP21 were unaltered compared to cells without Dox-treatment. (*Figure 5B* and *Figure 5—figure supplement 1*). These findings confirm that USP21 can remove ubiquitin from eS10 and uS10 in an activity-dependent manner. Similar to what was observed upon USP21 expression, OTUD3 expression resulted in substantially reduced eS10 ubiquitylation and complete ablation of uS10 ubiquitylation following UV treatment compared to uninduced cells (*Figure 5C* and *Figure 5—figure supplement 1*). This reduction in eS10 and uS10 ubiquitylation was activity dependent as induction of inactive OTUD3 did not alter either eS10 or uS10 ubiquitylation upon UV treatment (*Figure 5D* and *Figure 5—figure supplement 1*). It was surprising that overexpression of the inactive versions of USP21 or OTUD3 did not result in a dominant negative phenotype with enhanced eS10 or uS10 ubiquitylation at steady state or reduced demodification kinetics following UV exposure. This observation suggests that the Dubs do not compete for the same binding surface on the ribosome despite their ability to demodify the same RRub sites.

## OTUD3 or USP21 loss-of-function reduces stall readthrough and extends eS10 ubiquitylation following RQC activation

To determine if USP21 or OTUD3 loss of function impacted RRub or RQC activity, we generated USP21 and OTUD3 knockout cell lines and USP21/OTUD3 double knockout cell lines using genome engineering approaches (*Figure 6—figure supplement 1A*). Two separate knockout clones for both USP21 and OTUD3 displayed reduced poly(A)-mediated stall readthrough which is consistent with our demonstration that overexpression of OTUD3 or USP21 enhances stall readthrough (*Figure 6A*). Combined loss of USP21 and OTUD3 did not further reduce stall readthrough compared to individual knockouts suggesting that the Dubs may be acting at distinct points within the RQC pathway (*Figure 6A*). We then evaluated the kinetics of eS10 demodification following UV-induced RQC activation in parental and knockout cell lines. Consistent with previous results, eS10 ubiquitylation was rapidly induced following UV exposure and was fully demodified to pre-exposure levels by 16 hr in parental 293T cells (*Figure 6B* and *Figure 6—figure supplement 1B*). USP21 or OTUD3 knockout cells displayed reduced demodification kinetics following UV exposure with eS10 ubiquitylation persisting up to 24 hr in OTUD3 knockout cells (*Figure 6B* and *Figure 6—figure supplement 1B*). OTUD3 and USP21 double knockout cells also displayed sustained eS10 ubiquitylation following UV exposure compared to parental controls (*Figure 6B* and *Figure 6—figure supplement 1B*). While a delay in eS10 demodification was observed in the knockout cells, the extent of eS10 ubiquitylation is clearly reduced compared to peak levels in all knockout cells indicating that other Dubs can compensate for the loss of USP21 or OTUD3 and that further redundancy exists in the pathway.

It is notable that we failed to observe an RQC phenotype upon siRNA-mediated knockdown of USP21, OTUD3, or combinations of candidate Dubs (*Figure 6—figure supplement 1C–F*). We also could not detect reduced stall readthrough upon knockdown of 24 additional Dubs that were not represented in our overexpression library (*Figure 6—figure supplement 1G*). These results demonstrate that complete genetic ablation of OTUD3 or USP21 expression is required to observe defects with RQC and RRub demodification following RQC activation.

## OTUD3 preferentially demodifies RQC RRub sites and is present within ribosome enriched fractions

Stressors that induce the integrated stress response (ISR) also induce RRub events on uS3 and uS5 that function in a distinct manner compared to eS10 or uS10 RQC ubiquitylation events (*Higgins et al., 2015*). To examine if OTUD3 or USP21 act specifically on eS10 and uS10 RQC RRub events or act more broadly on ubiquitylated ribosomes we utilized the OTUD3 and USP21 inducible cell lines. After induced Dub expression, cells were treated with DTT or harringtonine (HTN) which

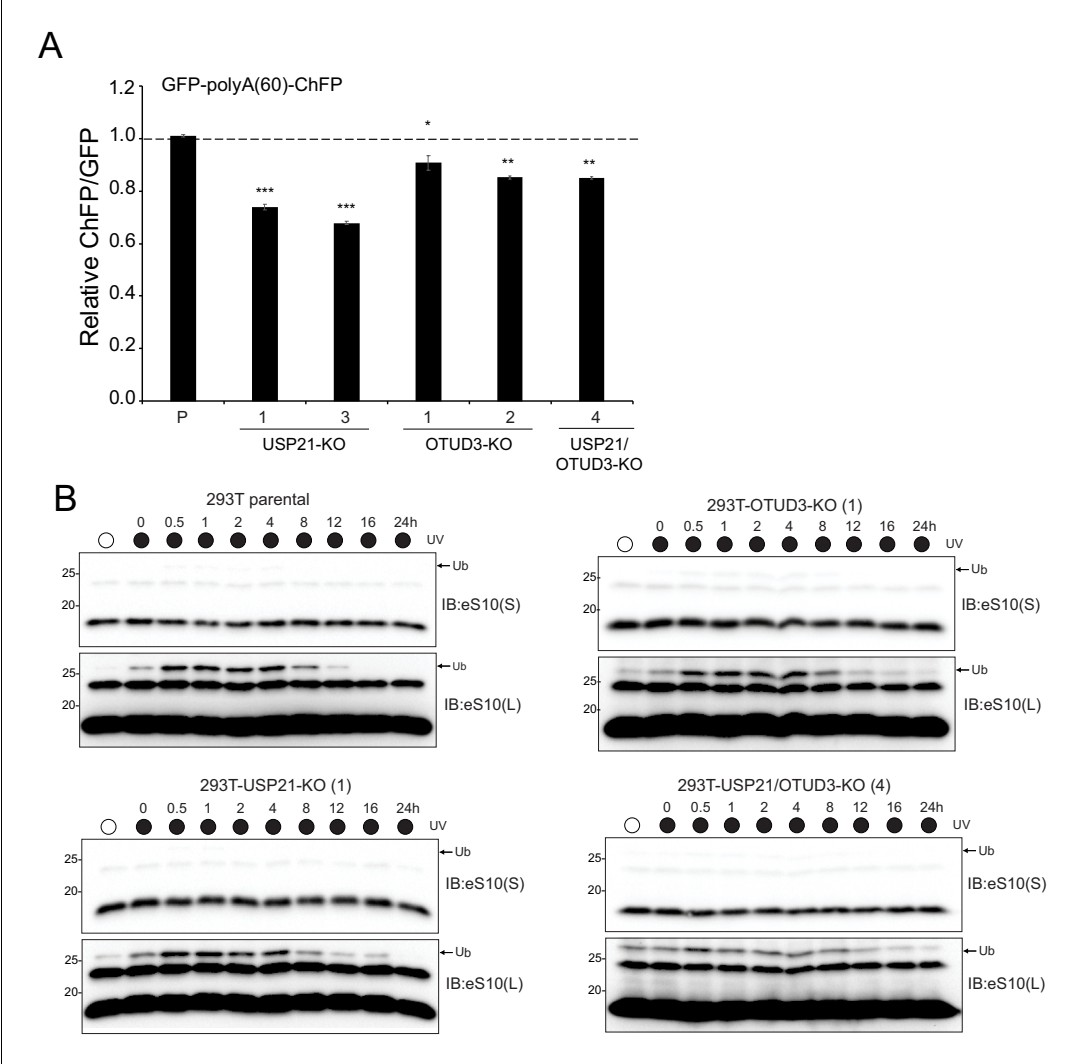

**Figure 6.** Loss of USP21 or OTUD3 expression results in enhanced ribosome stalling on poly(A) sequences and delayed eS10 ubiquitylation following RQC activation. (**A**) Parental 293 T cells (P), USP21 knockout (KO) cells, OTUD3-KO cells and the combined double-KO cells were transfected with the poly(A)-stall reporter. Fluorescence intensities were measured by flow cytometry and the relative ChFP:GFP ratio is depicted. Numbers represent distinct knockout clones for OTUD3 or USP21. Error bars denote SEM for triplicate transfections. \*\*\*p<0.0001, \*\*p<0.001, \*p<0.05 using Student's t-test comparing the different KO clones to the parental control transfection. (**B**) Parental 293T cells, USP21-KO (clone 1), OTUD3-KO (clone 1), and USP21/OTUD3 double-KO (clone 4) cells were exposed to UV and allowed to recover for the indicated times. Whole-cell extracts were analyzed by SDS-PAGE and immunoblotted with the indicated antibody. The ubiquitin-modified ribosomal protein is indicated by the arrow. S and L denote short and long exposures (representative immunoblots shown (n = 2).

The online version of this article includes the following figure supplement(s) for figure 6:

**Figure supplement 1.** Knockdown of OTUD3 or USP21 does not result in enhanced resolution of poly(A)-induced RQC.

stimulates uS5 and uS3 RRub events in a distinct manner (*Higgins et al., 2015*). Interestingly, USP21 expression reduced both HTN and DTT-induced uS5 and uS3 ubiquitylation in an activity-dependent manner (*Figure 7A*). However, OTUD3 expression reduced uS5 ubiquitylation, albeit to a lesser extent than UPS21 expression, and had no impact on HTN or DTT-induced uS3 ubiquitylation (*Figure 7B*). These results indicate that USP21 expression has a larger impact on all RRub events examined and that OTUD3 primarily demodifies ZNF598-catalyzed eS10 and uS10 ubiquitylation events.

Given the ability of USP21 and OTUD3 to remove RRub events, we examined if OTUD3 or USP21 associated with ribosomes. We treated OTUD3 or USP21 inducible cells with ANS in the presence or

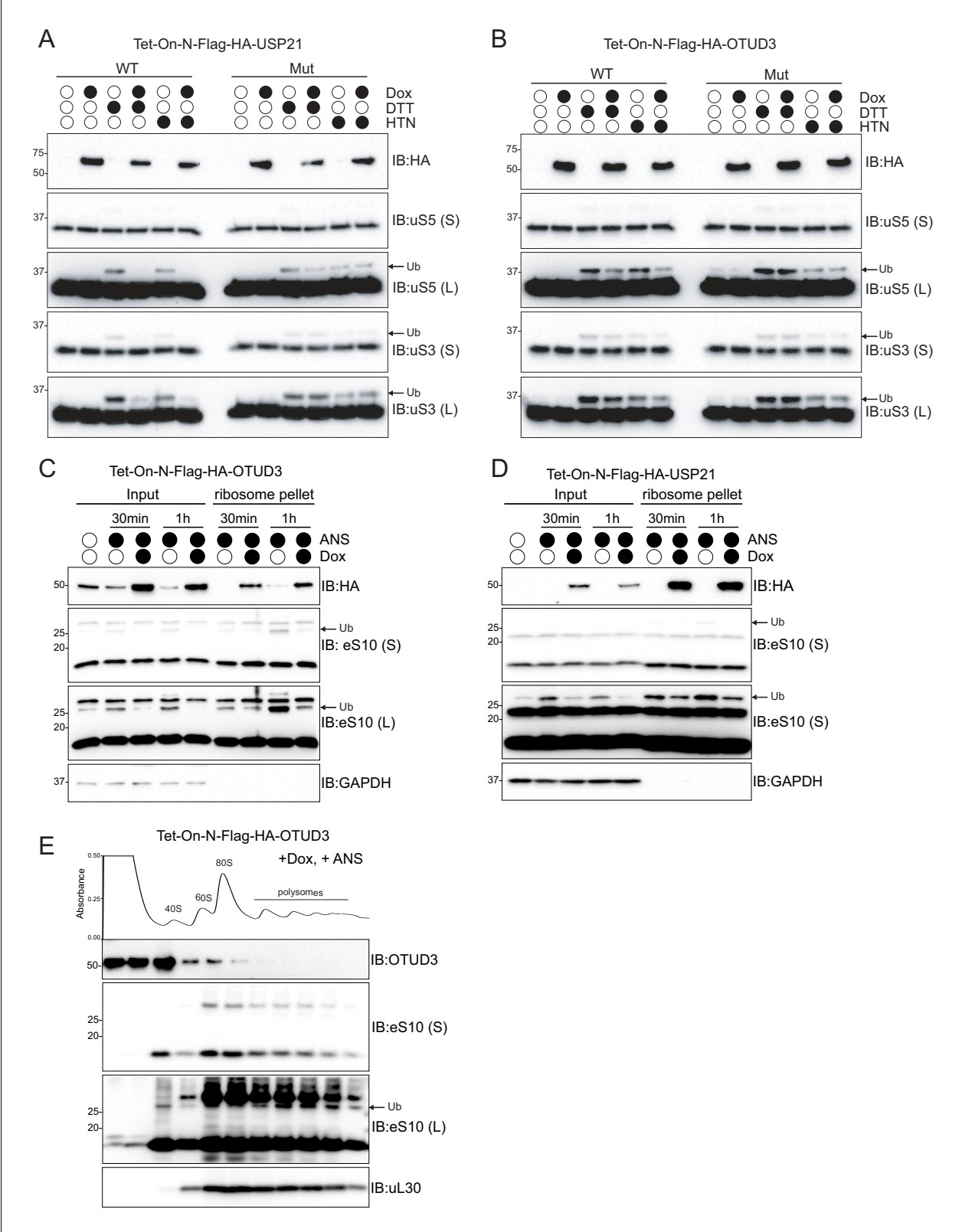

**Figure 7.** OTUD3 preferentially demodifies RQC RRub sites and is present within ribosome enriched fractions. (**A,B**) Cells with stable inducible expression of wild type USP21 (**A**), or OTUD3 (**B**) were induced or uninduced with doxycycline (2 μg/ml) for 16 hr and then treated with dithiothreitol (DTT, 5 mM) or harringtonine (HTN, 2 μg/mL). Whole-cell extracts were analyzed by SDS-PAGE and immunoblotted using the indicated antibodies. The ubiquitin-modified ribosomal protein is indicated by the arrow. S and L denote short and long exposures (n = 1). (**C,D**) Cells with stable inducible

*Figure 7 continued on next page*

*Figure 7 continued*

expression of wild type OTUD3 or USP21 were induced or uninduced and then treated with anisomycin (ANS 5 µg/ml) for the indicated times. Ribosomes were pelleted through a sucrose cushion and whole-cell extracts (input) and pelleted material were analyzed by SDS-PAGE and immunoblotted using the indicated antibodies. The ubiquitin-modified ribosomal protein is indicated by the arrow. S and L denote short and long exposures (n = 1). (E) Cells with stable inducible expression of wild type OTUD3 were induced for 16 hr followed by ANS (5 µg/ml) treatment for 1 hr. Whole-cell extracts were separated by sucrose density gradient centrifugation and fractions were collected. The UV absorbance across the fractions is depicted above the immunoblots. Individual fractions were TCA precipitated and analyzed by SDS-PAGE and immunoblotted using the indicated antibodies. The ubiquitin-modified ribosomal protein is indicated by the arrow. S and L denote short and long exposures (n = 1).

absence of Dox and then pelleted ribosomes by running whole cell extracts through a sucrose cushion. As expected, ANS induces eS10 ubiquitylation which is abrogated upon OTUD3 or USP21 expression (*Figure 7C,D*). Importantly, both OTUD3 and USP21 are present in the ribosome pellet fraction suggesting that OTUD3 and USP21 associates with ribosomes. To determine which ribosomal subpopulation is associated with OTUD3 upon RQC stimulation, we treated OTUD3 expressing cells with anisomycin and separated whole cell extracts over a linear sucrose gradient. Subsequent immunoblotting of gradient fractions revealed OTUD3 to be present in 40S containing fractions and largely absent from 80S and polysome containing fractions (*Figure 7E*). These results suggest that OTUD3 may preferentially demodify ubiquitylated 40S proteins that arise after subunit splitting and position OTUD3-mediated deubiquitylation as a putative late step during RQC events. Overall, our observations indicate OTUD3 and USP21 can demodify distinct sets of RRub events and regulate RQC activity.

## Discussion

Proteomics studies have revealed that a substantial portion of the ubiquitin-modified proteome may play a role in regulating cellular processes in a non-degradative capacity rather than targeting substrates for degradation (*Kim et al., 2011*). Several of these putative regulatory ubiquitylation events appear to be conserved, including many ribosomal monoubiquitylation events. Previous studies have established that conserved site-specific regulatory 40S ubiquitylation (RRub) is among the first signaling events required for proper RQC execution (*Garzia et al., 2017*; *Juszkiewicz and Hegde, 2017*; *Matsuo et al., 2017*; *Sundaramoorthy et al., 2017*). While the ubiquitin ligase that catalyzes the RQC-specific RRub events has been characterized in both mammals and yeast, whether RRub demodification was a critical step in ultimate resolution of RQC events and the identity of potential RRub Dubs remained unknown.

Here we identify two Dubs, USP21 and OTUD3, that contribute to the removal of ubiquitin from 40S ribosomal proteins. Overexpression of USP21 or OTUD3 results in enhanced readthrough of a poly(A) stall-inducing sequence. USP21 and OTUD3 have overlapping yet distinct ribosomal protein substrate specificities. The ubiquitin-specific proteases (USP) and the ovarian tumor (OTU) family make up the two largest Dub families. While USP Dubs are typically more promiscuous with regards to the types of polyubiquitin linkages they demodify (*Faesen et al., 2011*), OTU Dubs have been shown to exhibit ubiquitin-chain linkage specificity (*Mevissen et al., 2013*). Consistent with these observations, USP21 is more promiscuous than OTUD3 in that USP21 can hydrolyze all four tested RRub events while OTUD3 preferentially demodifies ZNF598-catazlyed eS10 and uS10 ubiquitylation events. These results establish that Dubs can play a regulatory role within the RQC pathway.

The factors that govern the regulation of these Dubs and when RRub events are removed within the RQC pathway remain unclear. We postulate two ways that Dubs may act as regulators of the ISR and RQC pathways. First, USP21 and OTUD3 may limit the activity of ZNF598 and other RRub ligases through direct antagonism to prevent spurious RRub. Unregulated 40S ubiquitylation could result in premature translational attenuation and destruction of properly processed mRNAs. Though plausible, the observed substochiometric ratio of OTUD3 and USP21 relative to ZNF598 suggests that Dubs may not directly limit ZNF598 activity. The low expression levels of USP21 and OTUD3 relative to ZNF598 may ensure that when progression of the ribosome is halted to a degree that requires RQC activity, the forward reaction is favored. A second possibility is that following 80S splitting, Dubs serve to strip the 40S of its ubiquitin in order to recycle unmodified 40S complexes which can reenter the translation cycle. Implicit in this model is that a ubiquitylated 40S may prevent or

reshape translation initiation events, a scenario which has not been directly established. More in-depth biochemical studies are required to identify the factors and mechanisms that regulate these reversible ribosomal regulatory ubiquitylation events, and the timing by which OTUD3 and USP21 exert their activity.

The mechanistic role of these regulatory ubiquitylation events remains obscure. One possibility is that ubiquitylation serves as a scaffold for targeting endo- or exoribonucleases responsible for downstream degradation of the mRNA. Recent work has shown that the endonuclease Cue2 is recruited to collided ribosomal complexes and is responsible for cleaving the mRNA within the A site (*D'Orazio et al., 2019*). With the unique interface formed by collided disomes and the position-ing of ubiquitylated eS10 and uS10, it is conceivable that Cue2 uses its two ubiquitin binding domains to latch onto the stalled complex. Another possibility is that ribosome ubiquitylation repre-sents a ubiquitin code that distinguishes ribosomes that are simply paused at a specific codon to allow for proper nascent chain folding or to relocalize translation from those that are terminally stalled due to an irreconcilable defect in the mRNA. A ribosomal ubiquitin code implies a possible order of operations and suggests that individual RRub events may not be occurring simultaneously, but rather in succession. Support for this model comes from the hierarchical relationship we observed among the different sets of RRub events, where the loss of eS10 ubiquitylation results in a reduction in uS10 modification (*Figure 2A*). This observation suggests that eS10 is the first ubiquity-lation event required for RQC initiation. Taken together, these results suggest that the combined modification of both eS10 and uS10 is required for robust resolution of stalled ribosomes. This observation is replicated for ISR stimulated uS3 and uS5 ubiquitylation where loss of the ability to ubiquitylate uS3 results in a complete ablation of uS5 modification which suggest that hierarchical RRub events may regulate translation beyond RQC function (*Figure 2D*).

Current models suggest that collided ribosomes trigger ZNF598 mediated eS10 and uS10 ubiquitylation (*Ikeuchi et al., 2019*; *Juszkiewicz et al., 2018*). In this case, it remains conceivable that the individual ubiquitylation events on eS10 and uS10 may not be taking place on the same ribosome, but rather occur on neighboring, collided ribosomes. For example, upon collision with the trailing ribosome, ZNF598 may mediate ubiquitylation of eS10 on the leading ribosome followed by ubiquitylation of uS10 on the trailing ribosome or vice versa. This could require a specific conforma-tion of ZNF598 in order to traverse both ribosomes, or following addition of the first ubiquitin the ligase moves upstream to the next site. Support for this model comes from studies in yeast that indi-cate ribosome collisions are critical events to initiate the no-go RNA decay pathway (*Simms et al., 2017*), and are required for upstream cleavage of the defective mRNA by the endonuclease Cue2 (*D'Orazio et al., 2019*; *Guydosh and Green, 2017*). Having shown that modification of both eS10 and uS10 are required to prevent readthrough of polyA-induced stalls, it is probable that the colli-sion with the leading ribosome triggers combinatorial ubiquitylation events that are required for recruitment of downstream RQC factors. Further biochemical analysis is needed to determine the cellular role of RRub events in recruitment of quality control factors and reshaping the translational landscape.

## Materials and methods

### Plasmids

The dual-fluorescence translation stall reporter plasmids were described previously (*Juszkiewicz and Hegde, 2017*; *Sundaramoorthy et al., 2017*). All Dub coding regions were cloned into Myc-tagging CMV expression vectors using Gateway cloning system (Invitrogen). QuickChange site-directed mutagenesis was done utilizing PCR-based approaches (primers 5' to 3': OTUD3-C76S, GGGGACGGCAATAGCTTGTTCAGAGC; OTUD1-C320S, CATTCCAGACGGCAACAGCCTCTACC-GAGCTG; OTUB2-C15S, GGGGATGGGAACAGCTTCTACAGGG; USP10-C424S, GATCAA TAAAGGGAACTGGAGCTACATTAATGCTACACTG; USP21-C250S, CCTGGGAAACACGAGCTTCC TGAATGC), followed by Dpn1 digestion of template DNA and transformation of the mutated plas-mids into TOP10 *E. coli* cells. Plasmids were confirmed by sequencing and screened for expression by immunoblotting.

## Cell lines, transfections and siRNA

All reagents utilized in this study can be found in *Supplementary file 1*: Key Resources Table.

HCT116 and HEK293 cells were grown in DMEM (high glucose, pyruvate and L-Glutamine) containing 10% fetal bovine serum (FBS) and 1% penicillin/streptomycin and maintained in a 5% $CO_2$ humidified incubator. Where indicated, cells were exposed to 0.02 J/cm$^2$ UV radiation using a Spectorlinker XL-1000 (Spectronics) before harvesting or treated with 5 mM DTT or 2 µg/mL Haringtonine for 4 hr before harvesting. Ansiomycin was used at a final concentration of 5 µg/ml.

HCT116 knock-in cells (uS10 K4R/K8R and eS10 K138R/K139R) were generated using CRISPR/Cas9 genome engineering approaches (Biocytogen). Individual clones were first validated by genomic sequencing. Cells containing the desired point mutation were selected for validation by immunoblotting. USP21 and OTUD3 knock out cell lines were generated in the 293T cell background using CRISPR/Cas9 genome engineering utilizing three individual guide RNAs (oligonucleotides 5′ to 3′: USP21: CAAGTATCGGTGGAGCCCGG, GGTAGCTTGGATCCCACTCG, TATGGAGCACGAGGATTCGA; OTUD3: CGGAATCGGCCGGAGTCTGG, CAACGCTTGAGGCGGACGCT, GCTCTTGGTGATCAATTGGA). 293T cells were transfected with the pSpCas9(BB)−2a-GFP plasmid containing individual guide RNAs using lipofectamine 2000. After 2 days GFP positive cells were single cell sorted on a BD FACSAria Fusion (BD BioSciences) cell sorter. Cells were validated for loss of USP21 and OTUD3 by immunoblotting.

Stable doxycycline inducible cell lines expressing Flag-HA-tagged proteins were generated using the Flp-In system (Thermo Fisher) through single locus integration, and Hygromycin selection. Flp-In T-Rex 293 cells were transfected with Flp-In expression vectors for the gene of interest (listed in resource table) using TransIT 293 transfection reagent (Mirus) according to manufacturer guidelines. Cells seeded at 60% confluency were transfected for 24 hr followed by selection of stable expression clones with 100 µg/mL Hygromycin. Protein expression was induced with 2 µg/mL doxycycline 16 hr prior to UV exposure, drug treatment, or harvesting.

All cellular transient transfections were performed using Lipofectamine 2000 (Thermo Fisher) and siRNA (see *Supplementary file 1*: Key Resources Table) knockdown transfections were performed using Lipofectamine RNAiMAX (Thermo Fisher) according to manufacturer guidelines.

## Dual-fluorescence translational stall reporter assay

All dual-fluorescent reporter plasmid cellular transfections were done using Lipofectamine 2000 according to manufacturer guidelines. Cellular ChFP and GFP fluorescence intensities for 10,000 individual events were measured 48 hr following transfection on a BD LSRFortessa X-20 cell analyzer (BD Biosciences). FACS data was analyzed using FlowJo (v10.4.1).

## Immunoblotting

Cell pellets were resuspended in denaturing lysis buffer (8 M urea, 50 mM Tris-Cl, pH 8.0, 75 mM NaCl, 1 mM β-glycerophosphate, 1 mM NaF, 1 mM NaV, 40 mM NEM and EDTA-free protease inhibitor cocktail (Roche Diagnostics)) and kept on ice throughout preparation. Cell lysates were sonicated for 10 s at output of 3 W with a membrane dismembrator model 100 (Fisher Scientific) with a microtip probe followed by centrifugation at 15,000 rpm at 4°C for 10 min. Supernatant protein concentrations were determined by BCA Protein assay (Thermo Fisher). Laemmli sample buffer with β-mercaptoethanol was added to cell lysates and heated at 95°C for 10 min. Lysates were resolved on 12 or 15% SDS-PAGE gels then transferred to PVDF membranes (Bio-Rad) using Bjerrum semi-dry transfer buffer (48 mM Tris Base, 39 mM Glycine-free acid, 0.0375% SDS, 20% MeOH, pH 9.2) and a semi-dry transfer apparatus (Bio-Rad Turbo Transfer) at 25V for 30 min. Immunoblots were blocked with 5% blotting grade nonfat dry milk (APEX Bioresearch) in TBST for 1 hr, followed by diluted primary antibody in 5% BSA over-night. Membranes were probed with anti-RPS2 (Cat# A303-794A, Bethyl); anti-RPS3 (Cat# A303-840A, Bethyl); anti-RPS10 (Cat# ab151550, Abcam) (antibody was used in *Figures 1E*, *3E*, *4E*, *5A–D* and *7C,E*, *Figure 6—figure supplement 1E, F*); anti-RPS10 (Cat# A6056, ABclonal, this was used for all other RPS10 (eS10) immunoblots); anti-RPS20 (Cat# ab133776, Abcam); anti-ZNF598 (Cat# HPA041760, Sigma); anti-OTUD3 (Cat# ab107646, Abcam); anti-USP21 (RRID:AB_10603227, Cat# HPA028397, Sigma); anti-USP21 (Cat# ab171028, Abcam) (antibody was used in *Figure 6—figure supplement 1D*; anti-phospho-eIF2α (Ser51)(D9G8) (Cat# 3398S, Cell Signaling Tech); anti-c-Myc (9E10) (Cat# sc-40, Santa Cruz); anti-ubiquitin (Cat#

MAB1510, EMD Millipore); anti-HA (Cat# MMS-101P, Biolegend); anti-tubulin (Cat# 3873S, Cell Signaling Tech); anti-Rabbit IgG, HRP (Cat# W4011, Promega); or anti-Mouse IgG, HRP (Cat# W4021, Promega). Immunoblots were developed with Clarity Western ECL Substrate (Bio-Rad) and imaged on a Bio-Rad Chemi-Doc XRS+ system. Imagelab (Bio-Rad) software was used to process all blots with final images prepared in Adobe Illustrator.

## Sucrose cushion
Cells were lysed in 600 ul of (50 mM Hepes, pH 7.4, 100 mM KAc, 2.5 mM MgAc2, 0.5% Triton X-100, 1 mM DTT, 1 mM NaF, 1 mM NaV, 40 mM NEM and EDTA-free protease inhibitor cocktail) buffer and 500 ul of whole cell extracts were pelleted over a 0.5 M sucrose cushion (500 ul) by spinning whole cell lysate in a TLA120.2 rotor at 100,000 rpm at 4℃ for 35 min. Pelleted material was resuspended in Laemmli sample buffer with β-mercaptoethanol and heated at 95℃ for 10 min followed by standard immunoblotting.

## Sucrose gradient
Cells were lysed in 600 ul of (500 mM Tris, pH 7.8, 1.5M NaCl, 150 mM MgCl2, 0.5% NP-40, 150 ug/ml cycloheximide, 8 U/ml SUPERase In RNAse inhibitor, 1 mM NaF, 1 mM NaV, 40 mM NEM and EDTA-free protease inhibitor cocktail) lysis buffer followed by sonication and clarification of lysate at 15,000 rpm at 4℃ for 10 min. 500 ul of whole cell extract was fractionated over a 10–50% sucrose gradient (prepared on Gradient Master 108 (Biocomp): 1 min 38 s, 81.5 degrees, 18 rpm) spinning at 41,000 rpm at 4℃ for 2 hr in an SW41i rotor. 1 ml fractions were collected using a PGFip piston gradient fractionator (Biocomp). Protein fractions were precipitated overnight at 4℃ with 10% Trichloroacetic acid (TCA) followed by three washes with ice-cold acetone. Pellets were dried in Vacufuge plus (Eppendorf) at room temperature for 5 min then resuspended in Laemmli sample buffer with β-mercaptoethanol and heated at 95℃ for 10 min followed by standard immunoblotting.

## Quantification and statistical analysis
All FACS-based assays, plasmid transfections and siRNA transfections were performed in triplicate (n = 3) as biologically distinct samples. The mean ChFP:GFP ratio and SEM were calculated and compared to K20-reporter transfection alone, parental cell type or control siRNA knockdown. Immunoblot quantification of the relative ubiquitin modification was calculated by normalization of the individual Ub intensities for each time point to that of the no UV control. Significance (p value) was calculated using an unpaired two-tailed Student's t test using GraphPad Prism 7.0.

## Acknowledgements
We thank the Goldrath lab (UCSD) for providing assistance on all FACS-based experiments. We thank Julie Monda for providing a critical reading of this manuscript. This work was supported by a UCSD Cell and Molecular Genetics Training Program (T32GM007240) and a National Science Foundation Graduate Research Fellowship (DGE-1650112) (DMG), and the NIH (DP2-GM119132 and PGM085764) (EJB).

## Additional information

### Funding

| Funder | Grant reference number | Author |
| --- | --- | --- |
| National Institutes of Health | DP2-GM119132 | Eric J Bennett |
| National Institutes of Health | PGM085764 | Eric J Bennett |
| National Science Foundation | DGE-1650112 | Danielle M Garshott |
| University of California, San Diego | Cell and Molecular Genetics Training Program T32GM007240 | Danielle M Garshott |

The authors declare that the funders had no role in designing this study, data collection, or interpretation.

## Author contributions

Danielle M Garshott, Conceptualization, Investigation, Visualization, Methodology, Writing - original draft, Writing - review and editing; Elayanambi Sundaramoorthy, Marilyn Leonard, Investigation, Writing - review and editing; Eric J Bennett, Conceptualization, Supervision, Funding acquisition, Visualization, Methodology, Writing - original draft, Writing - review and editing

## Author ORCIDs

Danielle M Garshott 🔟 http://orcid.org/0000-0002-4357-1781
Elayanambi Sundaramoorthy 🔟 http://orcid.org/0000-0003-1256-9758
Eric J Bennett 🔟 https://orcid.org/0000-0002-1201-3314

## Decision letter and Author response

Decision letter https://doi.org/10.7554/eLife.54023.sa1
Author response https://doi.org/10.7554/eLife.54023.sa2

## Additional files

### Supplementary files

• Supplementary file 1. Key resources table.

• Transparent reporting form

### Data availability

No datasets were generated in this study.

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
