## [Decision Letter]

[Editors’ note: the authors submitted for reconsideration following the decision after peer review. What follows is the decision letter after the first round of review.]

Thank you for submitting your work entitled "Distinct regulatory ribosomal ubiquitylation events are reversible and hierarchically organized" for consideration by *eLife*. Your article has been reviewed by three peer reviewers, one of whom is a member of our Board of Reviewing Editors, and the evaluation has been overseen by a Senior Editor. The reviewers have opted to remain anonymous.

All three reviewers appreciated the content of the manuscript wherein an overexpression screen was used to identify two specific DUBs that increase readthrough on RQC reporters (like the E3 ligase ZNF598) and reverse ubiquitylation on four key ribosomal proteins implicated in RQC and the ISR. They also appreciated the identification of a hierarchy of ubiquitylation on these different proteins though its relevance remains unclear. However, while each of the reviewers appreciated the substantial amount of work involved and the development of an impressive set of tools for monitoring and defining the relevance of specific ubiquitylation events in RQC (and the ISR), all three reviewers were concerned by the absence of a loss-of-function phenotype associated either with RQC (as assessed by the reporters) or with deubiquitylation at the sites of interest. Following our discussion, we agreed that it is essential for publication in *eLife* that the deletion cell lines be constructed in order to rigorously evaluate the role of these DUBs in RQC and in site-specific ubiquitylation. While these are challenging experiments, it seems possible that the siRNA experiments with the DUBs have been unsuccessful in revealing phenotypes simply because they are enzymes and an incomplete knockdown is insufficient. We appreciate that these experiments may take a considerable amount of time (more than two months) and so per the instructions at *eLife* we are rejecting the manuscript at this stage (unless these experiments are underway and ready more quickly than anticipated). We hope that the comments of the reviewers (included below) will help you in revisions of the manuscript and in identifying a path forward for this work.

*Reviewer #1:*

In this work, Garshott et al. use immunoblotting and a dual fluorescence reporter assay coupled with pharmacological and genetic perturbations to identify 2 deubiquitylases (Dubs), Usp21 and Otud3, which they propose participate directly in ribosome quality control (RQC). While the field (and this lab in particular) has over the past few years clearly defined the r-protein targets (eS10 and uS10) of a key E3 ligase, ZNF598, in mediating the various events of RQC, and several r-protein targets (uS3 and uS5) that are ubiquitylated by the ISR, what is not clear is the fate of the ubiquitylated proteins. Here the authors argue that ubiquitylation is not terminal (i.e. ubiquitylated proteins are not targeted to the proteasome) but that the ubiquitylated proteins are eventually de-ubiquitylated through the activities of a key set of Dubs that in particular antagonize the ZNF598-mediated ubiquitylation steps and thus limit RQC activation. Additionally, the authors provide data to indicate that there is a hierarchy of ubiquitylation events, where uS10 and uS5 modifications depend on prior events at eS10 and uS3. Together, these data lead the authors to suggest a model whereby 40S ribosomal proteins, which become ubiquitylated upon ribosome stalling or activation of the integrated stress response, can be deubiquitylated following dissociation of the ribosomal subunits and thereby recycled for further rounds of translation.

While this study represents the accumulation of considerable amounts of data and interesting ideas, there are limitations to the study and its interpretation that must be addressed. The primary issue is that the major phenotypes observed throughout the manuscript are only seen as the result of overexpression, thus diminishing the strength of all conclusions since the ability to address specificity is limited. The absence of a knockdown phenotype is consistent with the possibility of redundancy (where the authors haven't identified all relevant Dubs) or with the possibility that the Dubs are not relevant to RQC under standard conditions. These concerns coupled with small differences throughout that rely heavily on statistics for significance weaken the overall conclusions. In fact, throughout the manuscript, asterisks are clearly meant to denote significance, and yet it was often not clear what the point of reference was; similarly, western blots are broadly utilized for data analysis and yet it was difficult to determine how many replicates were run. These substantive concerns should be addressed before publication of this manuscript.

Figure 1

These experiments present data from which the authors argue that ubiquitylation induced by either ANS, DTT, or UV is reversed independent of the proteasome (MG132) or new protein expression. It is difficult to assess the details of the experiment as concentrations of various reagents used were not indicated. Given that ANS is being used to stimulate RQC and to block overall protein synthesis, it is critical to know that sufficient levels are being used to fully block protein synthesis (perhaps with some metabolic labeling). Similarly, there was no positive control for MG132 function. These are critical controls for establishing a model where neither new protein synthesis nor proteasomal decay account for the loss of the ubiquitylation. Finally, the eS10 western patterns appear to be different in 1A and 1E – are different antibodies being used?

Figure 2

This is the key experiment in the manuscript where the authors overexpress the collection of Dubs and look for a readthrough phenotype with reporter, reasoning that loss of ZNF-mediated ubiquitylation might function in this way. This screen reveals a set of 5 candidate Dubs that promote readthrough on overexpression and which the authors validate by repeating this assay (this is not a secondary validation). Importantly, they show that two of these readthrough phenotypes appear to depend, at least in part, on a functional catalytic center (OTUD3 and USP21); it might be useful for the authors to discuss why the readthrough phenotype is only partly abrogated by the catalytically dead mutant even for the two favorite candidates. It would be helpful if the authors discussed the variable levels of expression for the different Dubs as they state their conclusions.

Figure 3

This figure does a good job establishing that OTUD3 and USP21 can act to deubiquitylate eS10 and uS10, and that readthrough reports directly on these events. Despite the fact that I was able to extract this conclusion, the figure is difficult. The use of shaded dots throughout the manuscript where the meaning of the shading changes in each figure is difficult – I recommend more detailed labeling to make things clearer.

Figure 4

This figure details some convincing experiments with some very nice tools (mutations in the relevant r-proteins) showing that there is a hierarchy of ubiquitylation. However, the readthrough assays that generally rely on very small magnitude differences did not reflect what might be anticipated if these ubiquitylation/deubiquitylation events are relevant to RQC. As one critical example, if eS10 ubiquitylation is needed for uS10 ubiquitylation, then Dub OE should have no effect in the eS10-KI background, and yet it did have an impact that wasn't much different than for the uS10-KI which does not abrogate eS10 ubiquitylation. The overall small differences, and the fact that there are essentially no differences with the catalytically dead variants, make these overall conclusions weak.

Figure 6—figure supplement 1

The authors make a strong effort to look for a deletion phenotype for the Dubs OTUD3 and USP21 and when that fails, to find another Dub that does matter. The negative data in this figure are important and concerning in terms of the likely relevance of these factors for RQC.

Figures 5/6

Because the authors see no phenotype of the deletion of the Dubs on RQC, they instead use a Dox-inducible system to document changes in the dynamics of ubiquitylation that result from overexpression of OTUD3 or USP21. Indeed, eS10 and uS10 ubiquitylation appear to be impacted by USP21 and OTUD3 overexpression. What would have been more compelling is to see the change in dynamics of ubiquitylation that results from knock down (or better, deletion) of these factors, which would better address the specificity of their action.

*Reviewer #2:*

This manuscript investigates four ribosome ubiquitination events (two involved in RQC, and two downstream of ISR), with an emphasis on the enzymes that reverse ubiquitination. Two primary new observations are reported. First, the authors show that overexpression of either of two DUBs (Usp21 and OTUD3) reverse ubiquitination at the four ribosome sites. Second, they find that ubiquitination is partially hierarchical, with modification of one site facilitating modification of another. The data themselves are convincing for the most part (see exceptions below) as are their interpretations. However, the biological significance of the observations is largely unclear. With the DUBs, there are as yet no consequences observed in loss-of-function attempts using RNAi. With the hierarchical ubiquitination, there is no insight yet into why this would be of value. There is no doubt that some of the tools generated in this study and the various observations will be of interest to the field. However, the absence of clear evidence that these DUBs are functionally relevant for RQC or ISR limits the overall scope of the study.

1) The conclusion that there are detectable effects in loss-of-function experiments is premature based on what is shown. It is crucial that a more thorough examination be performed before interpreting these negative data. First, it is prudent to directly examine ubiquitination levels of the 4 ribosomal proteins to see if they are changed when Usp21 and/or OTUD3 expression is ablated. The best experiment here would be a time course: treat with some ubiquitination inducing insult (e.g., ANS, DTT, or UV), withdraw the insult, and monitor the loss of ubiquitination. This is a more direct assay of their effect than the reporter assay, and the temporal dimension substantially increases the chances of seeing an effect than a steady-state measurement. Second, the use of knockdowns on proteins that act catalytically seems inadequate given that even very low levels of remaining protein might perform the job. I feel it is premature to interpret negative data without clean knockouts of Usp21, OTUD3, or both. I realize this will be time-consuming, but it should be relatively straightforward with current technology. I would strongly encourage making these cell lines and directly checking ubiquitination status as suggested above; if needed, I am happy to agree to extra revision time for this important experiment.

2) The pelleting assay for ribosome association of DUBs is too blunt to be of much use. I would recommend replacing it with a more careful fractionation, demonstration of some degree of specificity (e.g., testing at least one other DUB that does not modulate RRub on the ribosome), and analyzing endogenous OTUD3 (which seems to be detectable with available reagents). For example, a sucrose gradient separation of untreated cells versus those treated with ANS under collision conditions would be far more convincing and might also inform about whether the putative ribosome association is with all ribosomes, only polysomes, collision dependent, or only with separated subunits. Pelleting is undesirable because proteins pellet for lots of reasons unrelated to ribosome association including aggregation, association with other large structures, membrane association, sticking to plastic tubes, etc.

3) It is somewhat surprising that over-expression of a catalytically inactive DUB at 100-1000-fold above endogenous levels does not have dominant-negative effects. In other cases where a DUB is acting in a specific pathway, such dominant-negative effects are quite potent (see PMID 19818707 for an example). What is the authors' explanation for this? The only instance where there is even a hint of dominant-negative effects is eS10 ubiquitination with OTUD3-CS overexpression (Figure 5D). The absence of dominant-negative effects seems to favour a model in which there is no specific binding site for these DUBs (see above for my concerns regarding the evidence for an interaction via pelleting assays), and hence no real specificity to the deubiquitination reaction. In this view, the reason only some DUBs show an effect is simply because only some DUBs have the capacity to cleave the bond between substrate and the first ubiquitin (many DUBs selectively recognize the di-ubiquitin interface, often of particular linkage types). I feel this is a real concern in how the data are interpreted and the authors should be cautious in discussing the relevance of their observations. The absence of a dominant-negative effect is worth noting in the manuscript.

4) In the co-expression experiments in Figure 3A, it seems important to verify that ZNF598 expression is equal under conditions where the DUB is co-expressed versus not. Otherwise, the effect (or at least part of it) might be a consequence of simply affecting ZNF598 levels. Evidence that this is part of the explanation is seen in Figure 3D where ZNF598 levels are higher when expressed on its own versus co-expressed with a DUB. Because Figure 3A does not contribute much to the authors' argument, one option is to simply eliminate it. If they do include the figure, then the ZNF598 blot seems essential to a proper interpretation.

*Reviewer #3:*

Failure to remove defective mRNAs or nascent chains can lead to the accumulation of cytotoxic protein aggregates and proteotoxic stress. Recent studies demonstrated that the ubiquitination triggered the RQC response on ribosomes stalled on poly-lysine encoding mRNA reporter, indicating that the role of ribosome ubiquitination in quality control is conserved. In the mammal, the E3 ligase ZNF598 catalyzes regulatory ribosomal ubiquitylation of specific 40S ribosomal proteins required for downstream RQC events. ZNF598 ubiquitinates the ribosomal proteins eS10 at lysines K138 and K139 and uS10 at K4 and K8. In yeast, the ubiquitination of the ribosomal protein uS10 at K6 or K8 residues by the E3 ubiquitin ligase Hel2 is essential for RQC. In this study, the authors identify OTUD3 and USP21 as deubiquitylating enzymes that antagonize ZNF598-mediated 40S ubiquitylation and facilitate ribosomal deubiquitylation following RQC activation. Overexpression of either USP21 or OTUD3 enhances readthrough of stall-inducing sequences as compared to knock-in cells lacking individual ribosomal ubiquitylation sites suggesting that combinatorial ubiquitylation of RPS10 (eS10) and RPS20 (uS10) is required for optimal resolution of RQC events and that deubiquitylating enzymes can limit RQC activation.

Overall, this article is of immediate interest in the field and logically presented. The discovery of the deubiquitylating enzymes that antagonize ZNF598-mediated 40S ubiquitylation is significant. Overexpression of either USP21 or OTUD3 enhances readthrough of stall-inducing sequences. However, the quadruple knockdown of OTUB2, OTUD1, OTUD3, USP21 did not affect RQC (translation arrest by polyA sequence). Therefore, the relevance of the deubiquitylation of RPS10 and RPS20 by USP21 or OTUD3 in RQC is still unclear. To be published in *eLife*, it is mandatory to demonstrate the significant roles of the deubiquitylating enzymes in RQC or stress response to UV or DTT.

[Editors’ note: further revisions were suggested prior to acceptance, as described below.]

Thank you for resubmitting your work entitled "Distinct Regulatory Ribosomal Ubiquitylation events are reversible and hierarchically organized" for further consideration by *eLife*. Your revised article has been evaluated by Cynthia Wolberger as the Senior Editor, a Reviewing Editor and one reviewer.

The manuscript has been substantially improved and we are provisionally accepting it for publication in *eLife*. The authors made great efforts to address previous reviewer comments and the manuscript represents important work for the field on a challenging problem. As you will see in the attached comments from two of the previous reviewers, both were pleased to see that the new cell lines (lacking the DUBs of interest) had been constructed and were able to provide some additional (important) support for the model that these newly characterized DUBS (OTUD3 and USP21) contribute to RQC. Despite this enthusiasm, both reviewers felt that it should be mentioned that the data are not a "slam dunk". The effects on read-through appear are modest, albeit significant, and the reviewers recognize that this may reflect limitations of the assay; they recommend caution in discussing this result. The demodification assays were more difficult to see as a very strong piece of evidence for the model, and they recommend a more nuanced description of the result; Westerns are difficult to use for the evaluation of subtle effects, and I think it is safe to say that any effects observed here are subtle. Lastly, the biochemical sedimentation assay lacks a relevant negative control, thus limiting its contribution.

Reviewer #1:

In this 2nd submission of their paper, Garshott et al. seek to address a crucial question raised by all three reviewers in the first round of review – that is, is there a loss-of-function phenotype associated with the 2 deubiquitylases (Dubs) that they study? In the first submission, the authors performed siRNA knockdowns which demonstrated no identifiable phenotype associated with loss-of-function of OTUD3 and USP21. In the current submission, they generate multiple knockout lines of both proteins and use these to show that both single- and double-knockouts cause (subtle) decreases in frameshifting using their polyA stalling reporter, as well as a potential (subtle) increase in the persistence of es10 ubiquitylation following UV exposure. The results for the readthrough assay are shown to be significant and thus support their claims of the relevance of these DUBs for ribosome quality control (RQC). I am left unconvinced by the demodification data and the biochemistry indicating association with 40S subunits.

Reviewer 1’s last point:

This is the major experimental point addressed by this revision. It is impressive to see that the authors made multiple lines for both single knock outs and that these respond similarly (in both experiments). There is a single line shown for the double knock out which I presume is because these lines are hard to generate. The readthrough data looks reasonably convincing though it is not clear to me why the phenotype of the double KO would not be stronger than either single. The demodification data were harder to interpret. The reader is expected to assess extent of demodification in the absence of quantitation, and this visual assessment depends entirely on the exposure that the authors show – and they don't all look equal. To my eye, the most positive interpretation is that for the double KO, the level of ubiquitination is more even all the way across – less of a peak with UV treatment – and thus there is high background ubiquitination that is neither strongly increased with UV treatment nor removed following UV treatment. But overall, Figure 6B does not prevent a very compelling case to me.

Reviewer #2:

The authors have addressed my original comments. In particular, the analysis of de-ubiquitination in time courses and readthrough in reporter assays were performed in cells knocked out for the candidate DUBs. The results generally support the idea that these two DUBs are functionally involved. Identification and validation of DUBs that reverse key ubiquitin modifications in the RQC pathway is an important contribution to the field.

Point 2:

The sucrose sedimentation association of OTUD3 with 40S is not very convincing and importantly lacks a control with another DUB that plays no known role in RQC.

---

## [Author Response]

[Editors’ note: the authors resubmitted a revised version of the paper for consideration. What follows is the authors’ response to the first round of review.]

All three reviewers appreciated the content of the manuscript wherein an overexpression screen was used to identify two specific DUBs that increase readthrough on RQC reporters (like the E3 ligase ZNF598) and reverse ubiquitylation on four key ribosomal proteins implicated in RQC and the ISR. They also appreciated the identification of a hierarchy of ubiquitylation on these different proteins though its relevance remains unclear. However, while each of the reviewers appreciated the substantial amount of work involved and the development of an impressive set of tools for monitoring and defining the relevance of specific ubiquitylation events in RQC (and the ISR), all three reviewers were concerned by the absence of a loss-of-function phenotype associated either with RQC (as assessed by the reporters) or with deubiquitylation at the sites of interest. Following our discussion, we agreed that it is essential for publication in eLife that the deletion cell lines be constructed in order to rigorously evaluate the role of these DUBs in RQC and in site-specific ubiquitylation. While these are challenging experiments, it seems possible that the siRNA experiments with the DUBs have been unsuccessful in revealing phenotypes simply because they are enzymes and an incomplete knockdown is insufficient.

To address this primary concern, we have generated individual USP21 and OTUD3 knock out cell lines (in the 293T background) and a cell line in which both USP21 and OTUD3 are knocked out using CRISPR-based approaches. We have verified the loss of protein expression by western blotting and have utilized multiple clones of the individual knock outs to test the impact of Dub knock out on two cellular phenotypes. We now demonstrate that loss of USP21 or OTUD3 results in a reduction in ribosome readthrough of a poly(A)-induced stall using our reporter assay. Combined loss of OTUD3 and USP21 also results in reduced readthrough. These results agree with our observation that overexpression of either Dub results in enhanced readthrough of the poly(A) stall reporter. Indeed, the reviewers and editors were correct in suggesting that the siRNA-mediated knockdown did not result in sufficient loss of Dub expression to reveal a phenotype. We then examined the kinetics of eS10 demodification after UV-induced RQC activation in these knock out cell lines. Loss of OTUD3 or USP21 resulted in delayed demodification relative to parental controls indicating that the Dubs act to remove eS10 ubiquitylation following stall-inducing events. The combined loss of both OTUD3 and USP21 did not result in a complete loss of eS10 demodification suggesting that other Dubs can remove eS10 in the absence of the primary Dubs. However, these results clearly demonstrate a loss-of-function phenotype for OTUD3 and USP21 in the RQC pathway. These results are depicted in the revised Figure 6 and the results with siRNA-mediated knockdown have moved to Figure 6—figure supplement 1. We feel it is important to report the absence of a phenotype upon siRNA-mediated knockdown of USP21 or OTUD3 to guide other researchers working on the RQC pathway.

We feel we have successfully addressed the primary concern raised by the reviewers and have done so within the 2-month revision window. These revisions represent a substantial amount of rigorous and careful work by the primary author to satisfy our own concerns and those of the reviewers. We have also reorganized the manuscript to make it easier to follow the flow of the experiments and results. As the changes are substantial, we have not indicated each individual change to the figures in this rebuttal letter.

We appreciate that these experiments may take a considerable amount of time (more than two months) and so per the instructions at eLife we are rejecting the manuscript at this stage (unless these experiments are underway and ready more quickly than anticipated). We hope that the comments of the reviewers (included below) will help you in revisions of the manuscript and in identifying a path forward for this work.Reviewer #1:[…] While this study represents the accumulation of considerable amounts of data and interesting ideas, there are limitations to the study and its interpretation that must be addressed. The primary issue is that the major phenotypes observed throughout the manuscript are only seen as the result of overexpression, thus diminishing the strength of all conclusions since the ability to address specificity is limited. The absence of a knockdown phenotype is consistent with the possibility of redundancy (where the authors haven't identified all relevant Dubs) or with the possibility that the Dubs are not relevant to RQC under standard conditions. These concerns coupled with small differences throughout that rely heavily on statistics for significance weaken the overall conclusions. In fact, throughout the manuscript, asterisks are clearly meant to denote significance, and yet it was often not clear what the point of reference was; similarly, western blots are broadly utilized for data analysis and yet it was difficult to determine how many replicates were run. These substantive concerns should be addressed before publication of this manuscript.

We agree that the use of asterisks was not entirely clear throughout the manuscript. We have addressed this concern with more explicit details for which samples are being compared to one another in the figure legends. We have also simplified some of the panels to make the comparisons more obvious. All of the main conclusions, for example, overexpression of OTUD3 or USP21 results in reduced eS10 ubiquitylation, have been validated using multiple western blots across multiple experiments.

Figure 1These experiments present data from which the authors argue that ubiquitylation induced by either ANS, DTT, or UV is reversed independent of the proteasome (MG132) or new protein expression. It is difficult to assess the details of the experiment as concentrations of various reagents used were not indicated. Given that ANS is being used to stimulate RQC and to block overall protein synthesis, it is critical to know that sufficient levels are being used to fully block protein synthesis (perhaps with some metabolic labeling). Similarly, there was no positive control for MG132 function. These are critical controls for establishing a model where neither new protein synthesis nor proteasomal decay account for the loss of the ubiquitylation.

We have updated the manuscript to explicitly denote the concentrations of the individual inhibitors used. We have not performed the requested metabolic labeling experiment as the concentration of ANS used in our studies [5ug/ml; 18.8µM] is above the previously reported concentration [10µM] that was demonstrated to block 90% of protein synthesis after 15 min of treatment. (J Biol Chem. 1967 Jul 10;242(13):3226-33).

Finally, the eS10 western patterns appear to be different in 1A and 1E – are different antibodies being used?

The reviewer is correct in noting that we switched which eS10 antibody we used midway through our study. We apologize for this confusion and have edited the Materials and methods section to indicate when each antibody was used and in which figures.

Figure 2This is the key experiment in the manuscript where the authors overexpress the collection of Dubs and look for a readthrough phenotype with reporter, reasoning that loss of ZNF-mediated ubiquitylation might function in this way. This screen reveals a set of 5 candidate Dubs that promote readthrough on overexpression and which the authors validate by repeating this assay (this is not a secondary validation). Importantly, they show that two of these readthrough phenotypes appear to depend, at least in part, on a functional catalytic center (OTUD3 and USP21); it might be useful for the authors to discuss why the readthrough phenotype is only partly abrogated by the catalytically dead mutant even for the two favorite candidates. It would be helpful if the authors discussed the variable levels of expression for the different Dubs as they state their conclusions.

We have added a statement addressing these observations in the Results section.

Figure 3This figure does a good job establishing that OTUD3 and USP21 can act to deubiquitylate eS10 and uS10, and that readthrough reports directly on these events. Despite the fact that I was able to extract this conclusion, the figure is difficult. The use of shaded dots throughout the manuscript where the meaning of the shading changes in each figure is difficult – I recommend more detailed labeling to make things clearer.

We apologize for the confusion. The shaded (grey) dots indicate that a catalytically inactive mutant (either for ZNF598 or tested Dubs) was used and a solid (black) dot indicates the wild type version. We have revised the figures and the figure legends to make this more explicit.

Figure 4This figure details some convincing experiments with some very nice tools (mutations in the relevant r-proteins) showing that there is a hierarchy of ubiquitylation. However, the readthrough assays that generally rely on very small magnitude differences did not reflect what might be anticipated if these ubiquitylation/deubiquitylation events are relevant to RQC. As one critical example, if eS10 ubiquitylation is needed for uS10 ubiquitylation, then Dub OE should have no effect in the eS10-KI background, and yet it did have an impact that wasn't much different than for the uS10-KI which does not abrogate eS10 ubiquitylation. The overall small differences, and the fact that there are essentially no differences with the catalytically dead variants, make these overall conclusions weak.

We demonstrate that loss of eS10 ubiquitylation (K138RK139R mutant) robustly reduces uS10 ubiquitylation but does not completely abolish uS10 ubiquitylation. This observation argues that USP21 or OTUD3 overexpression removes the residual uS10 ubiquitylation resulting in enhanced read-through. However, given that USP21 or OTUD3 overexpression results in a small enhancement of stall readthrough in the ZNF598 knockout cell lines which have no uS10 or eS10 ubiquitylation indicates that there is a small non-RRub impact on the stall assay upon Dub overexpression. This same effect may explain the enhanced readthrough observed upon Dub overexpression in the eS10 and uS10 knock-in cell lines.

Figure 6—figure supplement 1The authors make a strong effort to look for a deletion phenotype for the Dubs OTUD3 and USP21 and when that fails, to find another Dub that does matter. The negative data in this figure are important and concerning in terms of the likely relevance of these factors for RQC.Figures 5/6Because the authors see no phenotype of the deletion of the Dubs on RQC, they instead use a Dox-inducible system to document changes in the dynamics of ubiquitylation that result from overexpression of OTUD3 or USP21. Indeed, eS10 and uS10 ubiquitylation appear to be impacted by USP21 and OTUD3 overexpression. What would have been more compelling is to see the change in dynamics of ubiquitylation that results from knock down (or better, deletion) of these factors, which would better address the specificity of their action.

As indicated in the introductory paragraph of the rebuttal letter, we have generated USP21 and OTUD3 knock out cell lines and have demonstrated that loss of either Dub results in reduced stall read-through and delayed eS10 demodification following UV exposure (see revised Figure 6).

Reviewer #2:This manuscript investigates four ribosome ubiquitination events (two involved in RQC, and two downstream of ISR), with an emphasis on the enzymes that reverse ubiquitination. Two primary new observations are reported. First, the authors show that overexpression of either of two DUBs (Usp21 and OTUD3) reverse ubiquitination at the four ribosome sites. Second, they find that ubiquitination is partially hierarchical, with modification of one site facilitating modification of another. The data themselves are convincing for the most part (see exceptions below) as are their interpretations. However, the biological significance of the observations is largely unclear. With the DUBs, there are as yet no consequences observed in loss-of-function attempts using RNAi. With the hierarchical ubiquitination, there is no insight yet into why this would be of value. There is no doubt that some of the tools generated in this study and the various observations will be of interest to the field. However, the absence of clear evidence that these DUBs are functionally relevant for RQC or ISR limits the overall scope of the study.1) The conclusion that there are detectable effects in loss-of-function experiments is premature based on what is shown. It is crucial that a more thorough examination be performed before interpreting these negative data. First, it is prudent to directly examine ubiquitination levels of the 4 ribosomal proteins to see if they are changed when Usp21 and/or OTUD3 expression is ablated. The best experiment here would be a time course: treat with some ubiquitination inducing insult (e.g., ANS, DTT, or UV), withdraw the insult, and monitor the loss of ubiquitination. This is a more direct assay of their effect than the reporter assay, and the temporal dimension substantially increases the chances of seeing an effect than a steady-state measurement. Second, the use of knockdowns on proteins that act catalytically seems inadequate given that even very low levels of remaining protein might perform the job. I feel it is premature to interpret negative data without clean knockouts of Usp21, OTUD3, or both. I realize this will be time-consuming, but it should be relatively straightforward with current technology. I would strongly encourage making these cell lines and directly checking ubiquitination status as suggested above; if needed, I am happy to agree to extra revision time for this important experiment.

We thank the reviewer for this suggestion. The reviewer was correct in noting that the lack of phenotype observed with siRNA-mediated knockdown resulted from residual remaining Dub. As indicated in the introductory paragraph of the rebuttal letter, we have generated USP21 and OTUD3 knock out cell lines as well as a combined knockout cell line and have demonstrated that loss of either Dub results in reduced stall read-through and delayed eS10 demodification following UV exposure (see revised Figure 6).

2) The pelleting assay for ribosome association of DUBs is too blunt to be of much use. I would recommend replacing it with a more careful fractionation, demonstration of some degree of specificity (e.g., testing at least one other DUB that does not modulate RRub on the ribosome), and analyzing endogenous OTUD3 (which seems to be detectable with available reagents). For example, a sucrose gradient separation of untreated cells versus those treated with ANS under collision conditions would be far more convincing and might also inform about whether the putative ribosome association is with all ribosomes, only polysomes, collision dependent, or only with separated subunits. Pelleting is undesirable because proteins pellet for lots of reasons unrelated to ribosome association including aggregation, association with other large structures, membrane association, sticking to plastic tubes, etc.

While the OTUD3 antibody works in whole cell extracts, the dilution of OTUD3 across the gradient fractions is sufficient to eliminate any detectable signal for endogenous OTUD3 in sucrose gradient fractions. In lieu of this experiment, we utilized our inducible OTUD3 expression cell line and demonstrate that overexpressed OTUD3 is present in 40S containing fractions and largely absent from 80S or polysome-containing fractions upon ANS treatment (revised Figure 7E). This data suggests that OTUD3 may be acting primarily on the 40S after subunit splitting.

3) It is somewhat surprising that over-expression of a catalytically inactive DUB at 100-1000-fold above endogenous levels does not have dominant-negative effects. In other cases where a DUB is acting in a specific pathway, such dominant-negative effects are quite potent (see PMID 19818707 for an example). What is the authors' explanation for this? The only instance where there is even a hint of dominant-negative effects is eS10 ubiquitination with OTUD3-CS overexpression (Figure 5D). The absence of dominant-negative effects seems to favour a model in which there is no specific binding site for these DUBs (see above for my concerns regarding the evidence for an interaction via pelleting assays), and hence no real specificity to the deubiquitination reaction. In this view, the reason only some DUBs show an effect is simply because only some DUBs have the capacity to cleave the bond between substrate and the first ubiquitin (many DUBs selectively recognize the di-ubiquitin interface, often of particular linkage types). I feel this is a real concern in how the data are interpreted and the authors should be cautious in discussing the relevance of their observations. The absence of a dominant-negative effect is worth noting in the manuscript.

We agree with the reviewer and we also expected to observe a dominant-negative phenotype upon overexpression of the catalytically inactive USP21 or OTUD3. The lack of a dominant negative phenotype suggests that there are multiple interaction surfaces on ribosome for the Dubs or the Dubs are acting at different points within the pathway. We were also quite concerned that any Dub would result in RRub demodification due to the inherent ability, in vitro, of Dubs to demodify substrates. While the reviewer is correct in noting that some Dubs, in particular OTU-containing dubs, bind to a di-ub interface and show chain linkage specificity in vitro, nearly all USP domain Dubs can cleave the substrate-linked ubiquitin. We provide evidence here that directly address these issues. We might expect that overexpression of ANY USP containing Dub would display a phenotype in the stall assay when overexpressed because it would be able to remove RRub. This was clearly not the case as overexpression of most Dubs did not result in a stall phenotype. Further, we overexpressed 10 Dubs and tested their ability to demodify ZNF598 catalyzed eS10 ubiquitylation and only Dubs whose overexpression enhanced stall read through in the original reporter screen were able to reduce ZNF598-mediated eS10 ubiquitylation (see Author response image 1). This result demonstrates that overexpression of any Dub does not lead to RRub demodification and indicates some specificity with regards to Dubs that can act on RRub sites. We have added a statement noting the absence of a dominant negative phenotype for the catalytically inactive dubs in the Results section.

**Author response image 1. respfig1:** Specific Dub overexpression is required to antagonize ZNF598-mediated eS10 ubiquitylation. Myc-tagged ZNF598 was expressed alone or in combination with the indicated myc-tagged Dubs and whole cell extracts were blotted for myc or eS10. The intensity of the ubiquitylated eS10 band (normalized to unmodified eS10 intensity) relative to ZNF598 expression alone is depicted under the immunoblots.

4) In the co-expression experiments in Figure 3A, it seems important to verify that ZNF598 expression is equal under conditions where the DUB is co-expressed versus not. Otherwise, the effect (or at least part of it) might be a consequence of simply affecting ZNF598 levels. Evidence that this is part of the explanation is seen in Figure 3D where ZNF598 levels are higher when expressed on its own versus co-expressed with a DUB. Because Figure 3A does not contribute much to the authors' argument, one option is to simply eliminate it. If they do include the figure, then the ZNF598 blot seems essential to a proper interpretation.

To address this issue, we performed an additional experiment where we further increased the levels of ZNF598 relative to USP21 and quantified the levels of exogenous ZNF598 and USP21. As detailed in Author response image 2, quantification of the exogenous USP21 and ZNF598 protein levels in manuscript Figure 3—figure supplement 2C revealed a decrease in myc-USP21 levels at the highest level of myc-ZNF598 plasmid expression (Author response image 2). However, we repeated this experiment with a larger dose escalation of myc-ZNF598 relative to myc-USP21 and did not observe a consistent decrease in exogenous USP21 expression as ZNF598 expression increased (Author response image 2). Further, we examined the levels of endogenous ZNF598 protein expression upon overexpression of USP21 or OTUD3 and knockdown of USP21 or OTUD3 and did not observe any alteration in endogenous ZNF598 expression (Author response image 2). Taken together, these results indicate that USP21 or OTUD3 do not impact ZNF598 expression levels in a meaningful way.

**Author response image 2. respfig2:** Analysis of ZNF598 protein levels upon Dub overexpression. (**A**) Quantification of the relative amount of Myc-tagged ZNF598 or USP21 from immunoblots in Figure 3—figure supplement 2C. (**B**) Whole-cell extracts from HCT116 ZNF598 knock-out (KO) cells transiently co-transfected with increasing amounts of plasmid DNA for either wild type ZNF598 or USP21. Numbers indicate the ratio of transfected DNA for each plasmid. Extracts were analyzed by SDS-PAGE and immunoblotted for the indicated antibodies. S and L denote short and long exposures, respectively. (**C**) Quantification of the relative amount of Myc-tagged ZNF598 or USP21 from immunoblots in panel B. (**D**) HCT116 cells were co-transfected with expression plasmids for wild type (black circles) USP21, or OTUD3 and their respective catalytically-inactive mutants (grey circles). Cells were exposed to UV and allowed to recover for 4h. Whole-cell extracts were analyzed by SDS-PAGE and immunoblotted using the indicated antibodies. (**E**) Quantification of the relative amount of endogenous ZNF598 from immunoblot in panel D. (**F**) 293T cells were transfected with either control siRNA oligos or siRNA oligos targeting OTUD3, USP21 or ZNF598 as indicated. Whole-cell extracts were analyzed by SDS-PAGE and immunoblotted using the indicated antibodies. (**G**) Quantification of the relative amount of endogenous ZNF598 from immunoblot in panel F.

Reviewer #3:[…] Overall, this article is of immediate interest in the field and logically presented. The discovery of the deubiquitylating enzymes that antagonize ZNF598-mediated 40S ubiquitylation is significant. Overexpression of either USP21 or OTUD3 enhances readthrough of stall-inducing sequences. However, the quadruple knockdown of OTUB2, OTUD1, OTUD3, USP21 did not affect RQC (translation arrest by polyA sequence). Therefore, the relevance of the deubiquitylation of RPS10 and RPS20 by USP21 or OTUD3 in RQC is still unclear. To be published in eLife, it is mandatory to demonstrate the significant roles of the deubiquitylating enzymes in RQC or stress response to UV or DTT.

We thank the reviewer for their positive comments. While we have not documented a role for USP21 or OTUD3 in the ability to respond to UV or DTT, we now show that cells lacking USP21 or OTUD3 display reduced stall read-through and delayed eS10 demodification following UV exposure (see revised Figure 6).

[Editors’ note: what follows is the authors’ response to the second round of review.]

Thank you for resubmitting your work entitled "Distinct Regulatory Ribosomal Ubiquitylation events are reversible and hierarchically organized" for further consideration by eLife. Your revised article has been evaluated by Cynthia Wolberger as the Senior Editor, a Reviewing Editor and one reviewer.The manuscript has been substantially improved and we are provisionally accepting it for publication in eLife. The authors made great efforts to address previous reviewer comments and the manuscript represents important work for the field on a challenging problem. As you will see in the attached comments from two of the previous reviewers, both were pleased to see that the new cell lines (lacking the DUBs of interest) had been constructed and were able to provide some additional (important) support for the model that these newly characterized DUBS (OTUD3 and USP21) contribute to RQC. Despite this enthusiasm, both reviewers felt that it should be mentioned that the data are not a "slam dunk". The effects on read-through appear are modest, albeit significant, and the reviewers recognize that this may reflect limitations of the assay; they recommend caution in discussing this result. The demodification assays were more difficult to see as a very strong piece of evidence for the model, and they recommend a more nuanced description of the result; Westerns are difficult to use for the evaluation of subtle effects, and I think it is safe to say that any effects observed here are subtle. Lastly, the biochemical sedimentation assay lacks a relevant negative control, thus limiting its contribution.Reviewer #1:In this 2nd submission of their paper, Garshott et al. seek to address a crucial question raised by all three reviewers in the first round of review – that is, is there a loss-of-function phenotype associated with the 2 deubiquitylases (Dubs) that they study? In the first submission, the authors performed siRNA knockdowns which demonstrated no identifiable phenotype associated with loss-of-function of OTUD3 and USP21. In the current submission, they generate multiple knockout lines of both proteins and use these to show that both single- and double-knockouts cause (subtle) decreases in frameshifting using their polyA stalling reporter, as well as a potential (subtle) increase in the persistence of es10 ubiquitylation following UV exposure. The results for the readthrough assay are shown to be significant and thus support their claims of the relevance of these DUBs for ribosome quality control (RQC). I am left unconvinced by the demodification data and the biochemistry indicating association with 40S subunits.Reviewer 1’s last point:This is the major experimental point addressed by this revision. It is impressive to see that the authors made multiple lines for both single knock outs and that these respond similarly (in both experiments). There is a single line shown for the double knock out which I presume is because these lines are hard to generate. The readthrough data looks reasonably convincing though it is not clear to me why the phenotype of the double KO would not be stronger than either single. The demodification data were harder to interpret. The reader is expected to assess extent of demodification in the absence of quantitation, and this visual assessment depends entirely on the exposure that the authors show – and they don't all look equal. To my eye, the most positive interpretation is that for the double KO, the level of ubiquitination is more even all the way across – less of a peak with UV treatment – and thus there is high background ubiquitination that is neither strongly increased with UV treatment nor removed following UV treatment. But overall, Figure 6B does not prevent a very compelling case to me.

These comments center around the subtleness of the delay in ribosome deubiquitylation when performing the UV time course immunoblotting experiments in the parental and knockout (KO) cells. We agree that there is substantial demodification following UV exposure in the KO cells likely indicating the activity of other Dubs toward these modifications and articulated this in the previous version of the manuscript. We have addressed this issue in three ways. First, we further revised our discussion of these results to make it clear that the results clearly indicate that demodification is still taking place in the KO cells and that other Dubs, upon complete loss of the primary Dubs, can step in to demodify the ribosome following an RQC event. Second, we quantified (as suggested by the reviewer) the% eS10 ubiquitylation (not relative but absolute levels of eS10 modification) during the UV time course to make it easier to see the delay in eS10 demodification in the KO cells in the immunoblots. This is now included as Figure 6—figure supplement 1B. Here, it is obvious that eS10 ubiquitylation persists at 8, 12, and 16 hours post UV exposure compared to parental cells. Third, we repeated both the RQC reporter experiment and the UV time course experiment in the KO cells. The results replicated the data represented in the manuscript. We have included the repeated demodification immunoblots and quantification at the end of this letter. We would like to draw attention to the noticeable increase in eS10 steady-state ubiquitylation in the OTUD3 single KO and USP21/OTUD3 double KO lines. While the effects may not represent a complete block in eS10 demodification, loss of OTUD3 or USP21 results in reproducible effects on both the well-established single-cell RQC assay and eS10 demodification kinetics following UV exposure.

Reviewer #2:[…] Point 2:The sucrose sedimentation association of OTUD3 with 40S is not very convincing and importantly lacks a control with another DUB that plays no known role in RQC.

As the entirety of the samples for this experiment were utilized for the immunoblots shown in the figure, we did not have extra lysates to perform another immunoblot. Further, it is unclear which of the other 52 Dubs that did not score as RQC Dubs in our primary screen could be used as a suitable negative control. We felt that a random selection would not justify repeating this experiment. Also, as density centrifugation experiments merely measure the size distribution of assayed proteins, it is possible that the selected dub may associate with other cellular machinery that co-sediments at the 40S peak (or other peaks). We feel the density centrifugation experiment as depicted in the previous submission holds some value despite these limitations and have decided to include it in this revised version.